# Genetic diversity and phylogeography of *Phlebotomus argentipes* (Diptera: Psychodidae, Phlebotominae), using *COI* and *ND4* mitochondrial gene sequences

W. Methsala Madurangi Wedage[1], Iresha N. Harischandra[2,3], O. V. D. S. Jagathpriya Weerasena[4], B. G. D. N. K. De Silva[1,2,5] *

1 Center for Biotechnology, Department of Zoology, Faculty of Applied Sciences, University of Sri Jayewardenepura, Nugegoda, Sri Lanka, 2 Genetics and Molecular Biology Unit, Faculty of Applied Sciences, University of Sri Jayewardenepura, Nugegoda, Sri Lanka, 3 Vidya Sethu Foundation, Battaramulla, Sri Lanka, 4 Institute of Biochemistry, Molecular Biology and Biotechnology (IBMBB), University of Colombo, Colombo, Sri Lanka, 5 Sri Lanka Institute of Biotechnology (SLIBTEC), Homagama, Sri Lanka

* nissanka@sci.sjp.ac.lk

## Abstract

### Background

*Phlebotomus argentipes* complex is the primary vector for cutaneous leishmaniasis, a burgeoning health concern in contemporary Sri Lanka, where effective vector control is important for proper disease management. Understanding the genetic diversity of the *P. argentipes* population in Sri Lanka is vital before implementing a successful vector control program. Various studies have indicated that genetic divergence, caused by genetic drift or selection, can significantly influence the vector capacity of arthropod species. To devise innovative control strategies for *P. argentipes*, exploring genetic diversity and phylogeography can offer valuable insights into vector competence, key genetic trait transfer, and impact on disease epidemiology. The primary objective is to analyze the genetic diversity and phylogeography of the *P. argentipes* complex in Sri Lanka, based on two mitochondrial genomic regions in modern representatives of *P. argentipes* populations.

### Methodology

A total of 159 *P. argentipes* specimens were collected from five endemic areas of cutaneous leishmaniasis and identified morphologically. Two mitochondrial regions (Cytochrome c oxidase subunit I (*COI*) and NADH dehydrogenase subunit 4 (*ND4*) were amplified using the total DNA and subsequently sequenced. Partial sequences of those mitochondrial genes were utilized to analyze genetic diversity indices and to explore phylogenetic and phylogeographic relationships.

### Principal findings

Among five sampling locations, the highest genetic diversity for *COI* and *ND4* was observed in Hambantota (Hd—0.749, π—0.00417) and Medirigiriya (Hd—0.977, π—0.01055),

**Data Availability Statement:** All relevant data are within the manuscript and its Supporting information files.

**Funding:** This research was funded by the University Research Grant, University of Sri Jayewardenepura, grant number ASP/01/RE/SCI/2017/53, and the Centre for Biotechnology, Department of Zoology, Faculty of Applied Sciences, University of Sri Jayewardenepura, Sri Lanka. Snr. Prof. B.G.D.N.K. De Silva received the funding. The funders had no role in study design, data collection, analysis, the decision to publish, or manuscript preparation.

**Competing interests:** The authors have declared that no competing interests exist.

respectively. Phylogeographic analyses conducted using *COI* sequences and GenBank retrieved sequences demonstrated a significant divergence of *P. argentipes* haplotypes found in Sri Lanka. Results revealed that they have evolved from the Indian ancestral haplotype due to historical- geographical connections of the Indian subcontinent with Sri Lanka.

## Conclusions

Utilizing high-mutation-rate mitochondrial genes, such as *ND4*, can enhance the accuracy of genetic variability analysis *in P. argentipes* populations in Sri Lanka. The phylogeographical analysis of *COI* gene markers in this study provides insights into the historical geographical relationship between India and *P. argentipes* in Sri Lanka. Both *COI* and *ND4* genes exhibited consistent genetic homogeneity in *P. argentipes* in Sri Lanka, suggesting minimal impact on gene flow. This homogeneity also implies the potential for horizontal gene transfer across populations, facilitating the transmission of genes associated with traits like insecticide resistance. This dynamic undermines disease control efforts reliant on vector control strategies.

## Introduction

Genetic diversity is a fundamental element to represent evolution, and the process of adaptation and speciation are the foundation materials on which this genetic diversity depends [1]. High levels of genetic diversity in any population are generally considered a healthy characteristic, providing resistance to diseases, parasites, and predators, and to environmental changes [2]. The evolutionary trajectory of genetic diversity represents a critical phenomenon within the broader realm of the living community, reaching its peak of significance, particularly in comparison to the vector insect community [3]. Vector species, characterized by large populations and enhanced genetic diversity owing to mechanisms such as gene flow and outbreeding enhancement (heterosis), play pivotal roles in disease transmission dynamics [4]. However, the extent of genetic diversity within a vector population can significantly impact the efficacy of vector control and disease management practices, potentially fostering resistance to insecticides and sterile males [4, 5].

Leishmaniasis is an epidemic disease caused by protozoan parasites of the genus *Leishmania*, one of the vector-borne diseases that have recently received more attention. Cutaneous leishmaniasis, mucocutaneous leishmaniasis (MCL), and VL are the three primary clinical manifestations of the disease globally spread by phlebotomine sandfly vector [6, 7]. Leishmaniasis epidemiology in Sri Lanka gained prominence after 2001, becoming a notable, non-communicable disease [8]. The endemic nature of CL, with around 1991 annually reported cases in regions such as North Central Province (Anuradhapura District and Polonnaruwa District) and Southern Province (Matara and Hambantota districts), underscores the significance of effective disease management [9].

In this context, *Phlebotomus argentipes* Annandale and Brunetti (1908), one of the 1060 sandfly species globally, emerges as a main vector for VL and CL in Sri Lanka and India [7]. In Sri Lanka, *P. argentipes* is currently recognized as a species complex, and its distribution has been documented across all three climatic zones in the country [10, 11]. The distribution pattern of *P. argentipes* over the country highlights the epidemic risk of the disease, emphasizing the significance of vector control in effective disease management [8].

At present, there is no established method for controlling *P. argentipes* in Sri Lanka. The extensive utilization of insecticides like DDT and malathion, had been identified as effective control of other vectors, also it indirectly influences the development of insecticide-resistant *P. argentipes* populations [12]. As the *P. argentipes* vector species is well established in arid regions where malaria is prevalent, non-targeted control strategies are likely to have an effect on the development of insecticide resistance in these communities [11]. However, as a consequence of these factors, changes in the genetic composition of the *P. argentipes* community in Sri Lanka led to an increase in their genetic diversity. Several molecular studies of *P. argentipes* in Sri Lanka have been conducted and revealed the availability of higher genetic diversity [10, 11].

Compared to previous studies on other sandfly species, many factors, including altitude, interpopulation distances, habitat modifications, anthropogenically induced- landscape fragmentation, vegetation types, geographic barriers, and interactions with host communities collectively contribute to the distribution, abundance, and genetic diversity of phlebotomine sandflies in Sri Lanka [13–15]. Those factors intertwine to shape the genetic landscape of these vectors, impacting their vectorial capacity and disease transmission dynamics.

Molecular markers, including genomic DNA markers (microsatellites, Internal Transcribed Spacer 2 (ITS2) and mitochondrial markers [cytochrome b (*Cyt b*), *COI*, and *ND4*], have been used to determine the genetic diversity and phylogeography across diverse sandfly species such as *Phlebotomus papatasi*, *Phlebotomus perniciosus* and *Lutzomyia longipalpis* [16–22]. A recent study on the genetic diversity of *P. argentipes* in Sri Lanka highlighted distinct characteristics within local populations compared to other regions. The occurrence of gene flow among populations by existing a homogenized population has been suggested by identifying shared sequences (haplotypes) in the *COI* and *Cytb* genes. This raises concerns about the potential transfer of significant genes like insecticide resistance genes. Additional investigations, possibly using much more sensitive markers like microsatellites, are needed for confirmation of the exact picture of genetic diversity. A recent study also confirms the usefulness of *COI*, *Cytb* genes, and the ITS2 region as markers for assessing population structure [11].

While maternally inherited mitochondrial genes offer valuable insight into evolutionary history due to their clonal inheritance, absence of recombination, and higher mutation rate [23, 24], recent studies have utilized limited mitochondrial genes to explore the phylogeography of sandfly species [18, 25–30]. In the present study, extended fragments of the *COI* gene, exceeding the lengths employed in previous investigations of *P. argentipes*, the *ND4* fragments characterized by elevated mutation rates were utilized to explore the phylogeography and genetic diversity of *P. argentipes* in Sri Lanka.

Despite being the primary vector for CL in Sri Lanka, the existing scientific knowledge pertaining to the genetic variability of *P. argentipes* remains insufficient. To bridge this knowledge gap, the current study integrates molecular phylogenetic analysis, harnessing two mitochondrial DNA (mtDNA) genomic regions from modern representatives of *P. argentipes*, coupled with historical geographic data from Sri Lanka and India. The study aims to provide a robust phylogeographical interpretation, yielding insight into genetic structure and distribution. By fusing historical, geographical and behavioral factors, this analysis elucidates the intricate underpinning of genetic diversity within extant *P. argentipes* populations in Sri Lanka.

## Materials and methods

### Study area, sample collection, and identification

Sandflies were collected by using CDC light traps [31], sticky traps [32], and cattle-baited net traps [33] from five locations in Sri Lanka from March 2018 to March 2020. The sampling

locations were selected based on human CL disease prevalence in the country as per the reported patient numbers [9] (Fig 1, Table 1). Permission to collect mosquito samples was obtained from the respective Medical Officer of Health (MOH) of the study areas. Multiple

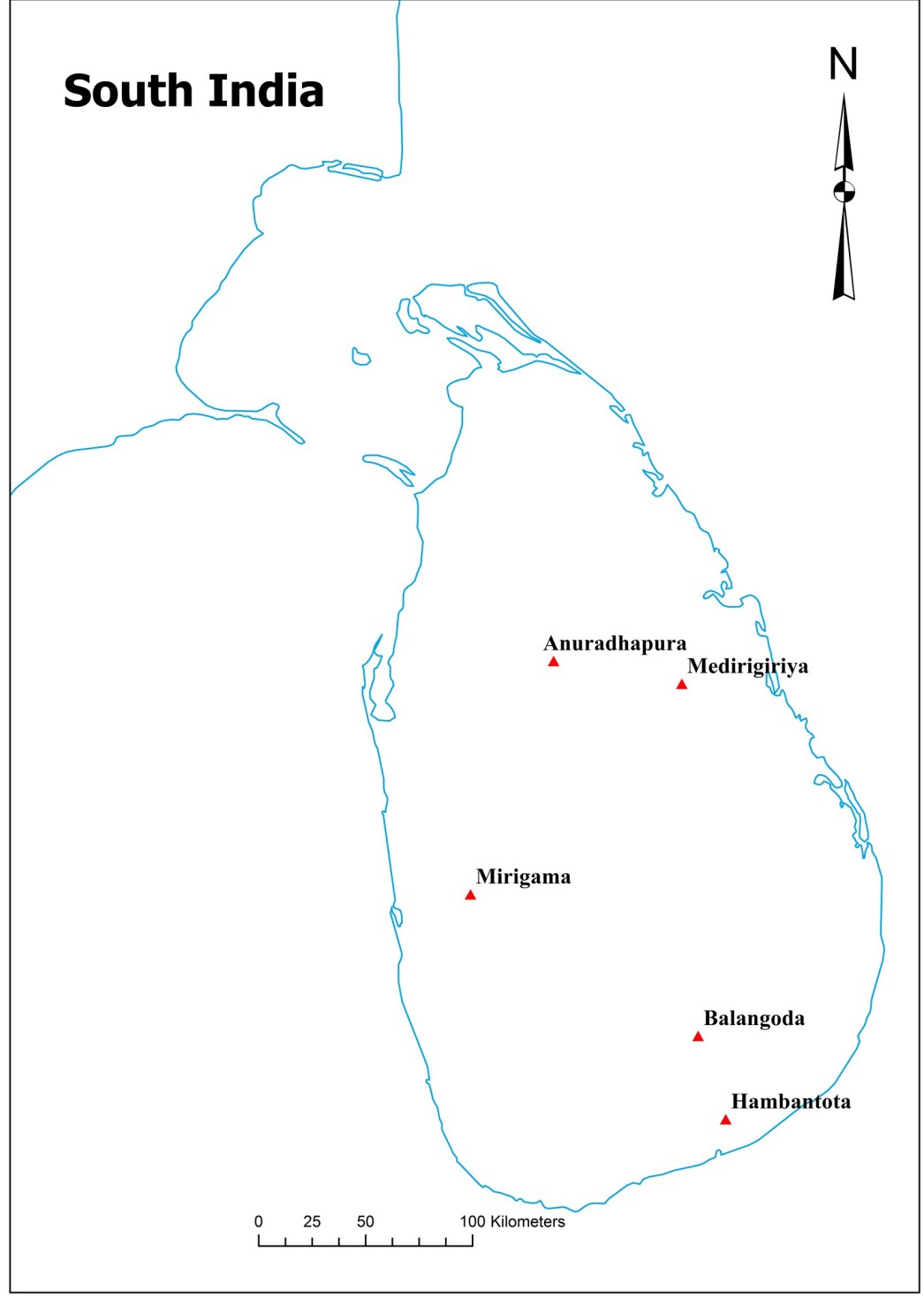

**Fig 1. Geographical distribution of *Phlebotomus argentipes* samples used in this study.**

**Table 1. GPS coordinates, accession numbers and the number of *Phlebotomus argentipes* specimens used for the analysis from five locations in Sri Lanka from March 2018 to March 2020.**

| Location ID | SampleCode | GPS coordinates (WGS 84) | | Number of *P. argentipes* specimens used for the analysis | Accession numbers of sequences deposited in GenBank database | |
| --- | --- | --- | --- | --- | --- | --- |
| | | Longitude | Latitude | | COI | ND4 |
| **Hambantota** | HAM | 6.313056˚ | 81.224167˚ | 41 | MZ882238-MZ882255 | MW002727, MW002737, MW002742, MW002756, MW002758- MW002765 |
| **Mirigama** | MIR | 7.258331˚ | 80.150831˚ | 31 | MZ882218-MZ882226 | MW002726, MW002730, MW002736, MW002754, MW002755, MW002757 |
| **Medirigiriya** | MED | 8.143417˚ | 81.039300˚ | 28 | MZ882227-MZ882237 | MW002728, MW002750 |
| | | | | | | MW002752, MW002766- MW002768 |
| **Balangoda** | BAL | 6.671942˚ | 81.023608˚ | 32 | MZ882204-MZ882217 | MW002725, MW002735, MW002746, MW002748, MW002749, MW002751, MW002753 |
| **Anuradhapura** | ANU | 8.239167˚ | 80.500000˚ | 27 | MZ882190-MZ882203 | MW002724, MW002724, MW002731- MW002734, MW002738, MW002739, MW002740, MW002741, MW002743- MW002745, MW002747 |

WGS84- World Geodetic System 1984.

collections at the same site were conducted. A total of 159 specimens of *P. argentipes* were used for the present analysis. The samples were stored in 70% ethanol at the field and immediately transferred to the -80 ˚C freezer at the Molecular Genetics Laboratory of the University of Sri Jayewardenepura for further analysis. *Phlebotomus argentipes* specimens were identified using the Magnus Microscopic Image projecting system (Magnus MIPS-India) and its live software based on the standard identification keys of Lewis [34] and Lane [35]. Identified sandfly specimens were preserved in 70% ethanol prior to extracting DNA.

## DNA extraction, PCR amplification and sequencing

Preserved samples were air dried for complete removal of ethanol. Then, the head parts of the specimens were preserved with 70% ethanol for further experiments, and the remaining body parts were used to extract DNA. According to the manufacturers' instructions, total genomic DNA was extracted from *P. argentipes* specimens using DNeasy Blood & Tissue kit (Qiagen, California, USA). The *COI* region was partially amplified [36] in a 25 μL reaction containing 1X PCR buffer, 1.5 mM MgCl$_2$, 0.2 mM dNTPs, 0.5 μM of each primer C1J (5′ -GGA GGA TTT GGA AAT TGA TTA GTT C- 3′) and C1N (5′ -CCC GGT AAA ATT AAA ATA TAA ACT TC- 3′), 1U of *Taq* DNA polymerase (Promega, USA) and 1–2 ng of template DNA. The cycle conditions [36] were modified with an additional step of 94 ˚C for 5 min at the beginning as the DNA preheat step. The *ND4* region was amplified [19] in a reaction volume of 25 μL contained 1X PCR buffer, 1.5 mM MgCl$_2$, 0.2 mM dNTPs, 0.5 μM of each primer ND4ar (5′- AA(A/G) GCT CAT GTT GAA GC- 3′) and ND4c (5′ -ATT TAA AGG (T/C)AA TCA ATG TAA- 3′), and 1–2 ng of template DNA. Cycle conditions were: 95 ˚C for 5 min, followed by 35 cycles of 95 ˚C for 30s, 48 ˚C for 45s, and 72 ˚C for 1 min, and final elongation at 72 ˚C for 10 min. Amplicons were purified and sequenced bi-directionally by the Sanger sequencing facility at Macrogen Inc, Korea.

## Genetic diversity of *P. argentipes* in Sri Lanka

Trace files of DNA sequences were assembled using DNA baser sequence assembler V 5.15, and the generated consensus sequences were searched over the GenBank database at NCBI using BLAST. The Open reading frames of the sequences were observed using the ORF finder

**Table 2.** *COI* and *ND4* datasets and haplotype counts based on the number of segregation sites.

| Analysis parameters | Cytochrome C Oxidase I | | NADH dehydrogenase subunit 4 (*ND4*) | Concatenated data set (*COI* + *ND4*) |
| --- | --- | --- | --- | --- |
| | *COI* study sequence alignment | *COI* regional sequence alignment | | |
| Number of Sequences used for the alignment | 148 | 167 | 133 | 121 |
| Length of the alignment (bp) | 455 | 343 | 598 | 1053 |
| Number of population sets included in the alignment | 5 | 7 | 5 | 5 |
| Number of considered segregation sites per alignment | 62 | 37 | 92 | 140 |
| Total number of haplotypes per alignment | 52 | 36 | 66 | 84 |

at NCBI [37]. Sequence sets were manually aligned with the ClustalW multiple alignment tool in MEGA 5.2 x, considering gaps as missing data. Genetic analyses were done through four alignments, and the sequence sets were indicated in Table 2. The four alignments analyzed in this article were *COI* study alignment (455bp), *COI* regional alignment (343bp), *ND4* study alignment (598bp) and concatenated alignment (1053bp).

Haplotypes for each alignment were generated using DnaSP v 5.10 [38]. All new haplotype sequences generated through this study for both mtDNA were deposited in GenBank, and the accession numbers were mentioned in Table 1. The *COI* and *ND4* sequences were aligned at 1,427 bp– 2,960 bp and 8,162 bp– 9,494 bp regions, respectively, against the reference sequence (KR349298.1) of *P. papatasi* complete mitochondrial genome [39] as no *P. argentipes* reference sequences available in databases.

Intra-population genetic diversity based on *COI* and *ND4* sequences was analyzed separately using the number of mutations, segregation sites, pairwise differences, haplotype number, haplotype diversity, nucleotide diversity, and nucleotide polymorphism index by using DnaSP v 5.10 [40].

## Phylogeographic analysis—Haplotype networks and phylogenetic relationships

The unique haplotypes for four alignments (Table 2) were separately analyzed to construct a haplotype network to determine the interrelationship between haplotypes.

The unique haplotypes for the Asian region (regional analysis) were analyzed using our *COI* sequences and 19 *COI* sequences previously deposited in the GenBank repository from India (KC791432 to KC791437, HQ541166, HQ585366 to HQ585373) [41] and Sri Lanka (KT428789 to KT478792) [10] (Table 1, Fig 1).

The evolutionary relationship between specimens was evaluated by constructing independent median-joining haplotype networks using Network v 10.0 software [42] by assuming epsilon equal to 0 and a transition/ transversion rate of 1:1.

## Results

### Sand fly collection, morphological identification, and DNA sequencing

Among the 159 collected *P. argentipes* specimens, 148 *COI* sequences and 133 *ND4* sequences have been successfully sequenced. The concatenated alignment (1053bp) was obtained from 121 study sequences used in phylogeography analysis (Table 2).

## Genetic diversity of *Phlebotomus argentipes* in Sri Lanka based on *COI* and *ND4* gene sequences

Sizes of *COI* and *ND4* PCR products were 500 bp and 800 bp, respectively, and the study alignments were prepared by trimming the sequences to 455 bp and 598 bp, respectively. *In silico* translations revealed no premature stop codons, ensuring that the sequences were free of nuclear mitochondrial DNA sequences (NUMTs).

The alignment of obtained *COI* sequences was composed of 62 polymorphic positions (13.62%) (39 transitions and 23 transversions) with 35.20% GC content, while *ND4* sequences showed 92 polymorphic sites (5.21%) (57 transitions and 35 transversions) with 27.4% GC content (Table 3, S1 Table).

We identified 52 haplotypes from *COI* alignment and 66 haplotypes for *ND4* with a range of 12–21 haplotypes per site. According to the analyzed data, there is a low level of genetic diversity for both mtDNA regions *COI* and *ND4*, with the nucleotide diversity (π) range of 0.004–0.002 and 0.01–0.004, respectively (Table 3). The highest haplotype diversity (Hd) and nucleotide diversity (π) for both gene fragments were recorded from Medirigiriya (Hd- 0.977, π- 0.01055) and Hambantota (Hd- 0.749, π- 0.00417) respectively.

## Genetic diversity of *Phlebotomus argentipes* based on *COI* regional data set

GenBank retrieved *COI* sequences of *P. argentipes* (regional alignment) were compacted to 343 bp. The alignment consisted of 37 polymorphic sites (15 transitions and 22 transversions) with 4.66% nucleotide polymorphism and 34.6% GC content (Table 3, S1 Table). Thirty-six haplotypes were reported from 167 sequence alignments (Table 2).

## Phylogeographic relationships of *Phlebotomus argentipes* in Sri Lanka based on *COI* and *ND4* study sequences

**COI.** About 63.0% of total mutation types recorded in *COI* alignment generated during the present study showed transitions, and the most frequent transition was Thymine—Cytosine (T-C 41.26%) (S1 Table). The frequent transversions comprised of Adenine- Cytosine, and Guanine- Thymine. The Transition to transversion ratio was approximately 3:2. The most frequent haplotype, *HC1* was reported from all sampling locations and determined as the ancestral haplotype in Sri Lanka. Six haplotypes (*HC2*, *HC9*, *HC12*, *HC15*, *HC17* and *HC34*) have shared within 2–5 locations (Fig 2). Location-specific haplotypes were observed from all sampling locations; the highest private haplotype count was observed from Hambantota compared to the other four sampling locations [Anuradhapura (n = 10), Balangoda (n = 9), Mirigama (n = 5) Medirigiriya (n = 6) and Hambantota (n = 13)] (Fig 2, S1 Fig). All haplotypes have been separated into nine divergent haplogroups representing a set of closely related genetic variants (Fig 2, S2 Table) according to the median-joining network analysis where topology does not coincide with the geographic distribution of the sampling locations. The genetic associations observed between haplotypes, as deduced from the network analysis, exhibit a lack of direct alignment with the geographical provenance of the sampled specimens.

**ND4.** In *ND4* alignment, 62.0% of point mutations were observed as transitions while Adenine—Guanine (A-G) was the most frequent. The highest abundant transversion was Adenine—Thymine (A-T 18.69%) (S1 Table). The transition to transversion ratio was approximately 9:7. The most frequent haplotype, *HN1* was shared by all sampling locations and identified as the ancestral haplotype. Nine haplotypes (*HN2*, *HN6*, *HN8*, *HN10*, *HN12*, *HN16*, *HN19*, *HN20*, and *HN32*) were shared within 2–5 locations (Fig 3, S2 Fig).

**Table 3. Genetic diversity indices for *Phlebotomus argentipes* populations based on *COI* and *ND4* gene sequences.**

| Parameters | Analyzed Alignments | | | | | | | | | | | | | | | | |
| --- | --- | --- | --- | --- | --- | --- | --- | --- | --- | --- | --- | --- | --- | --- | --- | --- | --- |
| | Cytochrome Oxidase I | | | | | | | | | | | | NADH dehydrogenase subunit 4 (*ND4*) | | | | |
| | A | | | | | B | | | | | | | | | | | |
| | ANU | BAL | MIR | MED | HAM | ANU | BAL | MIR | MED | HAM | In | SL | ANU | BAL | MIR | MED | HAM |
| **Number of sequences used** | 25 | 31 | 28 | 24 | 40 | 25 | 31 | 28 | 24 | 40 | 13 | 6 | 27 | 29 | 21 | 25 | 31 |
| **G+C content at coding positions** | 0.352 | 0.352 | 0.352 | 0.352 | 0.352 | 0.347 | 0.347 | 0.347 | 0.346 | 0.346 | 0.344 | 0.346 | 0.274 | 0.274 | 0.274 | 0.274 | 0.274 |
| S | 19 | 19 | 13 | 15 | 27 | 7 | 14 | 8 | 9 | 10 | 3 | 4 | 20 | 23 | 22 | 57 | 38 |
| η | 19 | 19 | 13 | 16 | 29 | 7 | 14 | 8 | 10 | 11 | 3 | 4 | 20 | 24 | 23 | 62 | 42 |
| π | 0.00406 | 0.00348 | 0.00232 | 0.00354 | 0.00417 | 0.00163 | 0.00281 | 0.00167 | 0.00263 | 0.00174 | 0.00135 | 0.00622 | 0.00453 | 0.00464 | 0.00483 | 0.01055 | 0.00517 |
| K | 1.847 | 1.583 | 1.056 | 1.609 | 1.896 | 0.560 | 0.963 | 0.571 | 0.902 | 0.597 | 0.462 | 2.133 | 2.707 | 2.776 | 2.886 | 6.310 | 3.092 |
| h | 14 | 14 | 9 | 11 | 18 | 7 | 11 | 7 | 7 | 9 | 4 | 2 | 17 | 18 | 12 | 21 | 18 |
| Hd | 0.81 | 0.8 | 0.55 | 0.75 | 0.75 | 0.43 | 0.55 | 0.39 | 0.45 | 0.36 | 0.42 | 0.53 | 0.94 | 0.93 | 0.91 | 0.98 | 0.83 |
| **Number of segregation sites analyzed** | 19 | 19 | 13 | 14 | 25 | 7 | 14 | 8 | 8 | 9 | 3 | 4 | 20 | 22 | 21 | 52 | 34 |
| $K_1$ | 3.818 | | | | | 5.719 | | | | | | | 8.986 | | | | |
| $K_2$ | 4.174 | | | | | 5.225 | | | | | | | 4.633 | | | | |
| $R$ | 1.937 | | | | | 2.559 | | | | | | | 2.235 | | | | |

(A) *COI* study alignment and (B) *COI* regional alignment, (In) GenBank retrieved, India *COI* sequence set, (SL) GenBank retrieved, Sri Lanka previous study *COI* sequence set, S- Number of variable sites, η—Total number of mutation, π- Nucleotide diversity per site, k- Average number of nucleotide differences, h- Number of haplotypes, Hd- Haplotype diversity, $K_1$ and $K_2$ are the transition/transversion bias rate ratios of purines and pyrimidines respectively, $R$- overall transition/transversion bias.

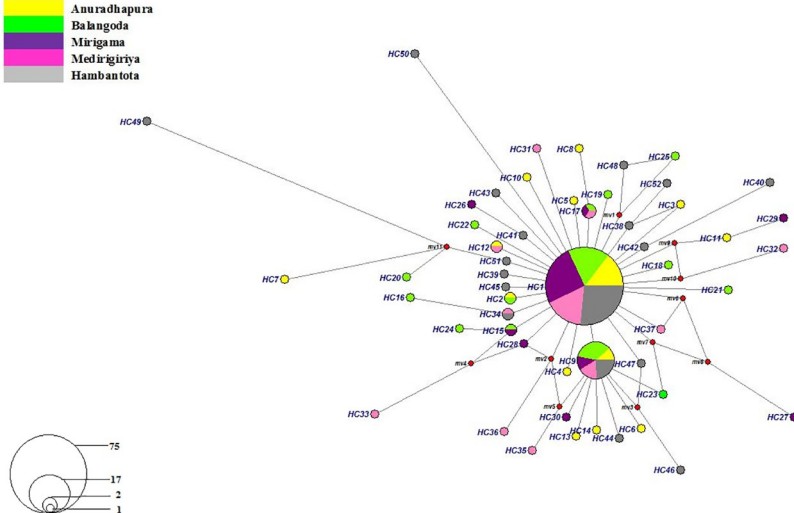

**Fig 2. Median-joining network for 148 *Phlebotomus argentipes COI* sequences of the current study.** Circle size indicates the haplotype frequency, and the circle color the geographical location. Haplotype labels are written next to the corresponding circles.

Private haplotypes were observed from all locations with respect to the *ND4* region of *P. argentipes* [Anuradhapura (n = 10), Balangoda (n = 11), Mirigama (n = 7), Medirigiriya (n = 15) and Hambantota (n = 12)] (Fig 3). All observed haplotypes have been categorized into three distinct haplogroups with the same common ancestor (*HN1*). The shared mutations have represented this and subsequently grouped haplotypes with comparable genetic characteristics into distinct clusters, thereby delineating the genetic diversity into discrete and discernible subgroups (S3 Table).

**Concatenated alignment.** According to the concatenated data set, four haplogroups (Table 4, S4 Table) were identified through median-joining network analysis.

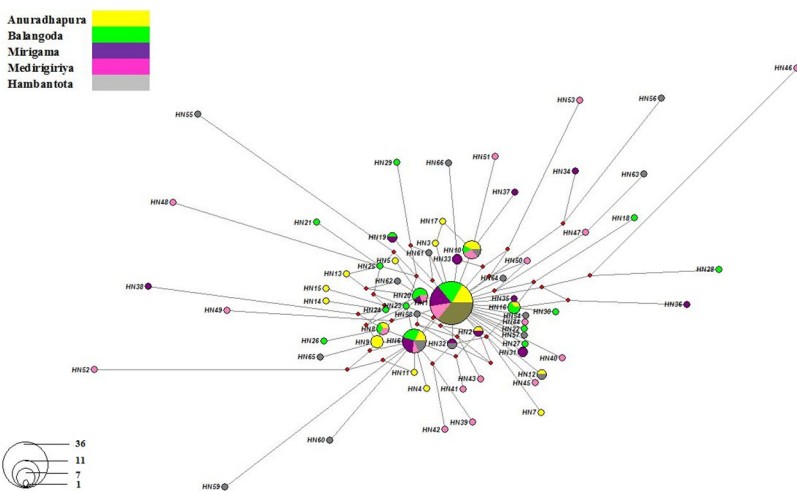

**Fig 3. Median-joining network for 133 *Phlebotomus argentipes* study *ND4* sequences.** Circle size and color indicate the frequency and geographical location of the haplotypes respectively. Haplotype labels are written next to the corresponding circles, and the red circles indicate median vectors.

**Table 4. Identified genetic clusters and haplogroups of *Phlebotomus argentipes* in Sri Lanka based on *ND4* and *COI* sequences and concatenated set of sequences.**

| Data set | Number of Haplogroups | Haplogroup IDs |
|---|---|---|
| *COI* study data set | 9 | I, II, III, IV, V, VI, VII, VIII, IX |
| Regional *COI* study data set | 5 | X, XI, XII, XIII, XIV |
| *ND4* study data set | 3 | XV, XVI, XVII |
| Concatenated (*COI* and *ND4*) data set | 4 | XVIII, XIX, XX, XXI |

## Phylogeographic relationships of *Phlebotomus argentipes COI* regional data set

The most frequent haplotype of *COI (HC1)* was not reported from GenBank-retrieved sequences from India (Fig 4A). However, *HC53*, *HC54*, *HC55* and *HC56* were found among sequences of Indian origin, and *HC56* showed an ancestral relationship with *HC1* (Fig 4B). Haplotypes of all analyzed *COI* sequences were grouped into five divergent haplogroups in regional *COI* alignment (Table 4, S5 Table), which showed relationships between the network topology and sampling locations. All previously studied *COI* sequences of *P. argentipes* were from 2012 to 2014. Only four sequences (KC791437, KC791435, KC791433, and KC791432) have consisted of *HC1* haplotype in regional analysis. The present study sequences in the regional analysis are based on *P. argentipes* samples examined and collected from 2018 to 2020. The haplotype network for the regional *COI* data set emphasizes an isolation discrepancy between recent and previously studied haplotypes based on the period susceptible to evolutionary changes (Fig 4).

## Discussion

There is ample research worldwide on sandfly genetic diversity [11, 14, 16, 42], but the genetic diversity and population genetic structure of *P. argentipes* have been marginally explored. This study was focused on analyzing the genetic diversity and phylogeography of *P. argentipes* based on *COI* and *ND4* partial mitochondrial sequences.

## Genetic diversity of *P. argentipes* in Sri Lanka

The nucleotide analysis of *P. argentipes* mtDNA revealed that *COI* and *ND4* possessed 64.80% and 72.60% A+T rich composition, respectively. It has been demonstrated that A+T rich

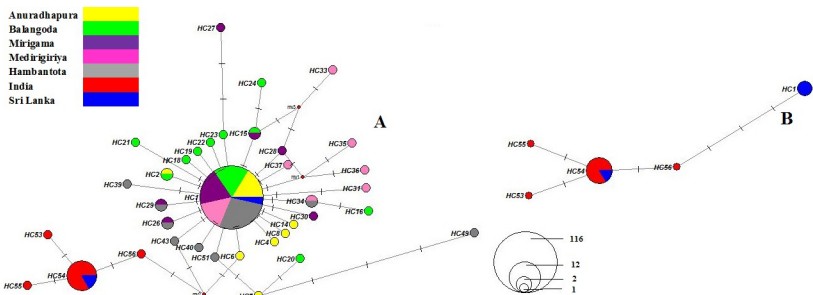

**Fig 4. Median-joining network for 167 *Phlebotomus argentipes COI* sequences including GenBank retrieved sequences.** Circle size and color indicate haplotype frequency and geographical location respectively. Haplotype labels are written next to the corresponding circles: (A) GenBank retrieved *COI* sequences/ regional alignment, (B) haplotypes derived from a previous study data set.

composition in a species gives rise to the diminution of synonymous position and will devastate the amino acid content, affecting the substitution percentage of the amino acid [43].

When considering the haplotype analysis, both gene regions manifested high-overall haplotype diversity (Hd) (Hd> 0.5), but relatively low nucleotide diversity (π) (π< 0.01). However, according to the regional *COI* analysis, relatively higher Hd and π values were shown by Sri Lankan (Hd- 0.53, π- 0.00622) samples rather than by Indian counterparts (Hd- 0.42, π- 0.00135). Due to many segregation sites, this has been attributed to the *COI* sequence length used for the developed haplotype analysis [44]. Genetic variability of populations based on mtDNA markers are classified into four categories as: (i) low Hd and low π (Hd < 0.5 and π < 0.005) (ii) high Hd and low π (Hd > 0.5 and π < 0.005) (iii) low Hd and high π (Hd < 0.5 and π > 0.005) (iv) high Hd and high π (Hd > 0.5 and π > 0.005) [45]. Across the various sampling locations examined within both genetic regions, the majority of observations in this study revealed high haplotype diversity (Hd) with low nucleotide diversity (π) (category ii). These trends are attributed to the limited number of nucleotide substitutions that have transpired among the identified haplotypes. Certain species of *Phlebotomus* genus have been found to exhibit similar patterns of high haplotype diversity and low nucleotide diversity in genetic diversity studies [28, 46, 47] that can be influenced by the factors such as population size, historical demographic events, and the presence of structuring forces like migration and selection [16].

One of the possibilities for such incidence could be population expansion after a period of a low effective population size followed by sudden population growth, which enhances the retention of new mutations, resulting from external factors associated with human activities and environmental conditions [48, 49]. The observed high haplotype diversity results from these dynamic population processes, where specific demographic events have led to the accumulation of genetic variation within the population [48, 49]. Lower genetic diversity within a vector population results from a past population bottleneck, followed by a decrease in effective population size due to natural environmental factors or human impacts. This bottleneck reduces genetic variability through random genetic drift, lowering overall genetic fitness [50]. Similar patterns are evident in other insect vectors [50], suggesting that *P. argentipes* may also undergo population reductions caused by intermittent bottlenecks triggered by fluctuations in host availability, breeding sites, or by other external factors. Conversely, population expansion occurs when a population rapidly grows after a period of small effective population size. Such expansion often follows a bottleneck, as surviving individuals reproduce and produce descendants [50]. New mutations are more likely to persist during population expansion, increasing genetic diversity. According to that, the current Sri Lankan *P. argentipes* population shows higher genetic diversity resulting from the above phenomena.

Only Medirigiriya and Hambantota sampling sites have been located in areas of higher epidemiological relevance [8]. It had been included in the fourth category of genetic variability (high Hd and high π) with respect to *ND4*. This indicates a high level of divergence between haplotypes. In this study, high haplotype diversity was found at both studied loci, with numerous unique haplotypes, suggesting either they have already been established long ago [51] or more sampling is needed to detect the current variation. The study of genetic markers such as the *ND4* gene has allowed for a finer resolution in the analysis of genetic variability within *P. argentipes* populations. Identifying numerous unique haplotypes, this research contributes to understanding the intricate genetic landscape of this sandfly species in CL endemic regions of Sri Lanka.

The transition/ transversion (ts/tv) ratio and overall ts/tv bias for *ND4* gene fragment (K1 = 8.986 for purines, K2 = 4.633 for pyrimidines, and R = 2.235) are higher than *COI* (K1 = 3.818 for purines, K2 = 4.174 for pyrimidines, R = 1.937) (Table 3). This is attributed to

post-mutation; the process occurs due to the high mutation rate of methylated cytosines to thymines in *ND4* than *COI* [52]. In addition to the high mutation rate and speed, *ND4* has many other advantages, including predefined marker systems, and PCR primers known to work for sandflies [19, 53, 54]. A higher substitution rate of the *ND4* gene than other DNA markers has facilitated an excellent marker for population genetics. The *ND4* gene offers distinct advantages in two specific scenarios. Firstly, it is well-suited for cases where multiple genetic mutations have accumulated over time. Secondly, it effectively addresses the propensity for certain genetic changes to occur more frequently than others during the evolutionary process [55]. The current findings concerning the *ND4* gene within the *P. argentipes* genome corroborate the validity of these attributes. In order to identify cryptic species, fast-evolving genes, such as *ND4*, may reach reciprocal monophyly between reproductively isolated populations.

Comparatively, most of the phylogeographic studies have been carried out based on the *ND4* than *COI* of mtDNA among these two markers based on the above facts [56].

## *COI* and *ND4* partial sequence-based haplotype distribution of *P. argentipes*

The *P. argentipes* population under study appeared to acquiesce to an extensive geographic transition by falling into nine haplogroups based on the *COI* region. The median-joining network analysis of the *COI* and *ND4* fragments identified the *HC1* and *HN1* as the ancestral haplotype, respectively. It presented and shared between five populations nucleated to the center of the radiating network. Other than the ancestral haplotype, the remaining haplotypes were differentiated from the ancestor by one to three mutational steps. Most of the mutated haplotypes were received from the Hambantota site, as the shared haplotypes were resident at the external nodes. The historical establishment of the *P. argentipes* population in the Sri Lankan context has been proven by the presence of many unique haplotypes for both genes, and the previous study on the *Cyt b* gene of *P. papatasi* in the Middle East and the Mediterranean region had endorsed the same phenomena [17, 18]. The *P. argentipes* population in Sri Lanka has dispersed over a mosaic terrain and geographic pattern with expansion events, could be a reason for multiple haplogroups in different sites with relevance to *COI* and *ND4* sequences as described by a previous study of *P. papatasi* in Morocco [20].

The patterns of haplogroups in all *COI* sequence analyses suggest that *P. argentipes* populations from widely separated and disjointed geographical areas are highly genetically differentiated, as shown by the differences in the frequencies of commonly occurring haplotypes. Two GenBank sequences, KC791436 and KC791434 from Punkuduthew and Delft islands of Northern Sri Lanka identified as geographically closer to the southern regions in India by analyzing KC791432 to KC791437, HQ541166, and HQ585366 to HQ585373 sequences from South India, Kerala, and Tamil Nadu regions. The close association of the haplotypes mentioned above of Southern Indian and Northern Sri Lankan populations suggests persistent genetic contact, the probability of gene flow before marine introgression in the Miocene period, and slow evolution following the tectonic events [57]. This could imply that genetic diversity or evolutionary history is not directly linked to geographic proximity. It might indicate factors like historical migrations, genetic exchange, or other evolutionary processes have influenced the genetic makeup of the populations in a way that does not follow simple geographic patterns. When considering the bionomics of the *P. argentipes* individuals, integration of the above factors, and the existence of Indian haplogroups in the northern regions of Sri Lanka due to ongoing gene flow/ migrations is a fact that could not explain or impracticable to explain by considering the bionomics of the sandfly. However, the ancestral relationships

among Indian and Sri Lankan *COI* haplogroups may have been established millions of years ago, based on the historical geography of the ancient Indian Peninsula with Sri Lanka [58].

## Conclusions

Utilizing genes with higher mutation rates, such as *ND4*, can offer enhanced efficacy in investigating genetic variability within a *P. argentipes* population, particularly when the analysis is confined to mitochondrial genes exclusively. This study substantiated the initial validation of the genetic homogeneity of the *P. argentipes* population in Sri Lanka, by employing the *ND4* gene as a pivotal marker.

Phylogeographical illustrations of *COI* gene markers provide insight into the historical-geographical relationships of the *P. argentipes* population in Sri Lanka and India. However, ancestral relationships are influenced by multiple factors, including gene flow, mutation rates, and population history, making it challenging to identify ancestral origins using these markers alone precisely.

The population of *P. argentipes* in Sri Lanka exhibits remarkable homogeneity, and this homogeneity remains consistent despite the diverse geographical nature of the country, indicating minimal influence on the gene flow. Consequently, there exists a potential for horizontal gene transfer through populations, facilitating the transfer of genes related to insecticide resistance and other traits. This phenomenon undermines the effectiveness of disease control measures through vector control strategies. However, extracting multiple microsatellite genotyping markers from the *P. argentipes* genome and conducting a genetic analysis with a more extensive sample size encompassing a broader geographical range could provide more significant substantiation for the assumption.

This work presents new insights towards understanding the genetic diversity of *P. argentipes* in CL endemic areas in Sri Lanka. Further analyses will be needed to elucidate how different vector populations shape the genetic variability and population structure of *P. argentipes*. Furthermore, the genetic structure of both vector and parasite populations should be studied to get a deeper understanding of the Eco epidemiology of Sri Lankan cutaneous leishmaniasis.

## Supporting information

**S1 Table. Parsimony informative sites for *COI* and *ND4* data sets of *Phlebotomus argentipes* and the mutation positions of alignments had been marked by using the reference mtDNA genome of *Phlebotomus papatasi* KR349298.1.** Parsimony informative sites for haplotypes were generated by using three sequence alignments [*COI* study alignment (455bp), *COI* regional alignment (343bp), and, *ND4* study alignment (598bp)] of *Phlebotomus argentipes*. The mutation positions in the alignments of *P. argentipes* were marked by using the reference mtDNA genome of *P. papatasi* KR349298.1.
(XLS)

**S2 Table. Identified haplogroups in *COI* study data set.** Haplogroups identified from the *COI* study alignment of *P. argentipes*, along with corresponding haplotypes within each group.
(DOC)

**S3 Table. Identified haplogroups in *ND4* study data set.** Haplogroups identified from the *ND4* study alignment of *P. argentipes*, along with corresponding haplotypes within each group.
(DOC)

**S4 Table. Identified haplogroups in concatenated study data set.** Haplogroups identified from the concatenated alignment of *P. argentipes*, along with corresponding haplotypes within

each group.
(DOC)

**S5 Table. Identified haplogroups in *COI* regional data set.** Haplogroups identified from the *COI* regional alignment of *P. argentipes*, along with corresponding haplotypes within each group.
(DOC)

**S6 Table. Haplotypes generated through each sequence alignment and the relevant haplotype IDs of each population.** Haplotypes presented in this table were identified from five sampling locations, each corresponding to four distinct alignments. The alignments include the *COI* study alignment (455 bp), *COI* regional alignment (343 bp), *ND4* study alignment (598 bp), and concatenated alignment (1053 bp). The haplotypes represent unique genetic variants observed in each alignment and sampling location combination.
(DOCX)

**S1 Fig. Median-joining network with mutation points, for 148 *Phlebotomus argentipes COI* sequences in the current study.** Circle size indicates the haplotype frequency, and the circle color the geographical location. Haplotype labels are written next to the corresponding circles, and the red circles indicate median vectors.
(TIF)

**S2 Fig. Median-joining network with mutation points, for 133 *Phlebotomus argentipes* study *ND4* sequences in the current study.** Circle size and color indicate the frequency and geographical location of the haplotypes respectively. Haplotype labels are written next to the corresponding circles, and the red circles indicate median vectors.
(TIF)

## Acknowledgments

The Epidemiology Unit, Ministry of Health, Sri Lanka, Anti- Malaria Campaign, and the regional public health officers of the sampling areas are acknowledged for providing leishmaniasis disease prevalence data and assisting the field works. Mr. Sanjaya Ranawaka, Survey Department of Sri Lanka, is acknowledged for supporting Cartography works. The collaboration, support, and expertise provided by laboratory colleagues at the Molecular Genetic Laboratory, Centre for Biotechnology, are gratefully acknowledged for their contributions throughout this research.

## Author Contributions

**Conceptualization:** W. Methsala Madurangi Wedage, Iresha N. Harischandra, B. G. D. N. K. De Silva.

**Data curation:** W. Methsala Madurangi Wedage.

**Formal analysis:** W. Methsala Madurangi Wedage, Iresha N. Harischandra.

**Funding acquisition:** B. G. D. N. K. De Silva.

**Investigation:** W. Methsala Madurangi Wedage, Iresha N. Harischandra.

**Methodology:** W. Methsala Madurangi Wedage, Iresha N. Harischandra, B. G. D. N. K. De Silva.

**Project administration:** W. Methsala Madurangi Wedage, B. G. D. N. K. De Silva.

**Resources:** B. G. D. N. K. De Silva.

**Software:** W. Methsala Madurangi Wedage.

**Supervision:** Iresha N. Harischandra, O. V. D. S. Jagathpriya Weerasena, B. G. D. N. K. De Silva.

**Validation:** W. Methsala Madurangi Wedage.

**Writing – original draft:** W. Methsala Madurangi Wedage.

**Writing – review & editing:** Iresha N. Harischandra, O. V. D. S. Jagathpriya Weerasena, B. G. D. N. K. De Silva.

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
