## [Decision Letter · Decision Letter 0]

1 Aug 2023

PONE-D-23-20755Genetic diversity and phylogeography of Phlebotomus argentipes (Diptera: Psychodidae), using COI and ND4 mitochondrial gene sequences.PLOS ONE

Dear Dr. De Silva,

Thank you for submitting your manuscript to PLOS ONE. After careful consideration, we feel that it has merit but does not fully meet PLOS ONE’s publication criteria as it currently stands. Therefore, we invite you to submit a revised version of the manuscript that addresses the points raised during the review process.

Please evaluate the comments made by the reviewers. The suggested changes and improvements are mandatory for future publication. Additionally, it is essential to have an assessment of the quality of the English, conducted by a native speaker of the language, to ensure understanding and clarity of the proposal.

We look forward to receiving your revised manuscript.

Kind regards,

Felipe Dutra-Rêgo, PhD

Academic Editor

PLOS ONE

5. We note that Figures 1 and 2 in your submission contain [map/satellite] images which may be copyrighted. All PLOS content is published under the Creative Commons Attribution License (CC BY 4.0), which means that the manuscript, images, and Supporting Information files will be freely available online, and any third party is permitted to access, download, copy, distribute, and use these materials in any way, even commercially, with proper attribution. For these reasons, we cannot publish previously copyrighted maps or satellite images created using proprietary data, such as Google software (Google Maps, Street View, and Earth). For more information, see our copyright guidelines: http://journals.plos.org/plosone/s/licenses-and-copyright.

1. You may seek permission from the original copyright holder of Figures 1 and 2 to publish the content specifically under the CC BY 4.0 license. 

Reviewers' comments:

Reviewer's Responses to Questions

**Comments to the Author**

1. Is the manuscript technically sound, and do the data support the conclusions?

Reviewer #1: Partly

Reviewer #2: Yes

Reviewer #3: Partly

2. Has the statistical analysis been performed appropriately and rigorously? 

Reviewer #1: No

Reviewer #2: Yes

Reviewer #3: N/A

3. Have the authors made all data underlying the findings in their manuscript fully available?

Reviewer #1: Yes

Reviewer #2: Yes

Reviewer #3: Yes

4. Is the manuscript presented in an intelligible fashion and written in standard English?

Reviewer #1: Yes

Reviewer #2: Yes

Reviewer #3: Yes

5. Review Comments to the Author

Reviewer #1: The authors analyzed the genetic diversity of several populations of Phlebotomus argentipes in Sri Lanka based on two well recognized mitochondrial markers. Despite not using nuclear gene markers, many specimens were sampled for mtDNA.

Although the results increase knowledge about this important sand fly species, some adjustments need to be made to improve the article:

i) The bibliography is out of date, some recent studies on the molecular taxonomy and populations genetics of sand flies were not included. For instance another publication on the same species and with similar objectives was recently published for Sri Lanka (https://doi.org/10.1371/journal.pone.0256819). The authors need to discuss their results in relation to this and other studies, highlighting the contributions of their findings in relation to what is already known about this species;

ii) As this is a phylogeography study, it is interesting that the authors provide a phylogenetic analysis of the analyzed sequences. This helps in interpreting the results and visualizing the clustering pattern and evolutionary relationship of the sequences. I recommend using the IQTree software, which has a user-friendly webserver (http://iqtree.cibiv.univie.ac.at/);

iii) Genetic structure indices (such as the Fst) can improve the understanding on the molecular divergence (or the lack of) between the analyzed populations.

Other minor corrections and suggestions are in the attached pdf file.

Reviewer #2: Reviewer #1: The manuscript "Genetic diversity and phylogeography of Phlebotomus argentipes (Diptera:

Psychodidae), using COI and ND4 mitochondrial gene sequences” provides relevant information on the

genetic diversity and phylogeopgraphic distribution of the sand fly Phlebotomus argentipes from Sri

Lanka. In addition, it contributes to new sequences of mitochondrial markers COI and ND4 for

Phlebotomus argentipes. However, some corrections/suggestions are necessary, and are detailed in the

attached file. I suggested that you do a language review, there are several errors and it is not very clear in some sentences. Despite this, I considered that the authors have valuable information that can be published

if they make improvements to their manuscript.

Title: Lines1-3: I suggest added Phlebotominae in the title: Genetic diversity and phylogeography of

Phlebotomus argentipes (Diptera: Psychodidae, Phlebotominae) using COI and ND4 mitochondrial gene

sequences

Delete the point at the end of the title.

Key words: Cutaneous Leishmaniasis; Sandfly; Phlebotomus argentipes, Sri Lanka; Haplotypes, phylogeny,

Do not use the same words as in the title. Find complementary words.

Abstract

Line 49: A total of 159 P. argentipes specimens, were collected and identified from five cutaneous

leishmaniasis endemic areas.

Lines 50: Cytochrome Oxidase subunit I

Introduction

The introduction needs to be more specific for the study area, and to the sand fly species to be analysed.

There is a lot of information that is not relevant to this job. Please provide specific information

(background) on the topic you are going to develop.

Line 87: Out of 1,060 (Please update the number of species) sand fly species in the world, Phlebotomus

argentipes Annandale and Brunetti (year???) is one of the major vectors for Visceral Leishmaniasis (VL) and

Cutaneous Leishmaniasis (CL) in Sri Lanka???(or where?? specify).

I share this reference where the number of species worldwide is updated: Galati E, Rodrigues B. A review of

historical Phlebotominae taxonomy (Diptera: Psychodidae). Neotrop Entomol. 2023; 1–21.

doi:10.1007/s13744-023-01030-8

Line 89: Being CL the most common form, annually reporting 0.7-1.2 90 million cases worldwide [4], and

caused by nearly 20 species of obligate intracellular protozoa Leishmania sp. (Euglenozoa: Trypanosomatidae)

[5].

I agree that cutaneous leishmaniasis is the most common clinical form worldwide. But this sentence is not so

relevant in this part of the manuscript. I suggest that you to rewrite this paragraph.

Lines 91-95: In Sri Lanka, around 1,991 annual CL cases have been recorded mainly in endemic areas of the

North Central Province (Anuradhapura District and Polonnaruwa District) and Southern Province (Matara and

Hambantota districts) in the country (Fig. 1) [6-7]

Please change Figure to Fig throughout the text, considering the guidelines of the journal.

Lines 98-99: are believed to be high in the country. What do you mean with “are believed”? I suggest you be

more specific. Please only use facts (references) not assumptions.

Line 101: host communities, host species turnover??. Do you mean to blood meal preferences with this?

Line 101: vectorial capacity and competence. Do you considered the vectorial competence too??

Lines 103-105: I don’t understand why you include a map with the incidence of cutaneous leishmaniasis if

your manuscript focuses on the genetic diversity of sand fly Phlebotomus argentipes? Does the distribution of

Phlebotomus argentipes coincides with the sites where the cases have been reported? In this case is better

including the distribution of the sand fly.

I suggest that this map of cutaneous leishmaniasis cases send as supplementary information. Only

include a map indicating the sites where you collected for your study (in this case the Figure 2 should be the

Figure 1).

Line 103: Fig 1. The distribution of…

Lines 106-107: add the complete name of the genes since it is the first time they appear.

[microsatellites, the Internal Transcribed Spacer 2 (ITS2)] and mitochondrial markers [cytochrome b (cytb),

cytochrome oxidase subunit 1 (COI) and NADH Dehydrogenase 4 (ND4)] have been used [11-17], to

determine the genetic diversity and phylogeography of different sand fly species such as P. papatasi, P.

perniciosus and Lu. Longipalpis.

Lines 111-112: Thus, mitochondrial genes have been used in evolutionary studies of vector species including

phlebotomine sandflies. (It is necessary including a reference).

Lines 112-114: However, most studies have used few mitochondrial genes to estimate the phylogeography of

sand fly species.

Lines 114-118: In the present study, extended fragments of the COI gene, exceeding the lengths employed in

previous investigations of P. argentipes, meanwhile the ND4 fragments characterized by elevated mutation

rates, were utilized to explore the phylogeography and genetic diversity of P.argentipes in Sri Lanka.

Since in the previous paragraph I suggested putting the full name of the genes, in this paragraph

you can abbreviated.

Line 130: Material and Methods

Line 132: In those study areas there were no other species of sandflies?

Line 132: randomly??, Do you mean that during the three years of the sampling, you always placed the traps in

different places? Please explain why were the traps not placed in the same place during sampling? What would

be the explanation for the low abundance of this species during the three years of sampling? Since is a vector, I

consider that for a sampling of three years, 159 specimens are very few specimens. If you collected more

specimens but only used 159 for the molecular analysis, you should specify it.

Lines 133-1334: from five localities in Sri Lanka from March 2018 to March 2020 (Table 1).

Line 135: (Fig 2, Table 1).

Line 138: for further analysis. Phlebotomus argentipes specimens…

Line 140: I suggest that in Table 1 you also add the Genbank accession numbers generated in this study for

COI and NAD4 genes.

Line 144: Fig 2. Geographic distribution of Phlebotomus argentipes samples analysed. The sampling locations

of this study are marked with a triangle, and the GenBank sequences retrieved for mitochondrial DNA

fragments from Sri Lanka with a square, and those from India with a circle.

I suggest that you make the letter and symbols bigger for better visualization of the figure.

Line 149: How did you perform the morphological identification if you used the complete body for extraction

and the head for other experiments? For the morphological identification of Phlebotomus argentipes it is not

necessary to visualize the cibarium or the spermatheca? Please justify how you made the morphological

identification and if you made assemblies, what technique you followed and where the specimens are kept?

Line 160: Since the information that is in Table 1 is displayed in Figure 2, I consider it appropriate to join

table 1 and 2 and only keep the relevant information, to avoid the information being repetitive and confusing.

Only the data that you generated should be included in the table. The data that you downloaded from GenBank

it is written in the paragraph (Line184-186), it is not necessary to repeat the information.

Lines 173-174: The information in Table 3 can be described in the paragraph. It is not necessary the table to

describe this information. Please remove the table.

Line 177: specify what software you used to perform this procedure.

Lines 181-182: I do not understand, what you mean with MZ882190 to MZ882255. Please be more specific,

because it's confusing this way. Please do not repeat the information, is better if you refer to the table or figure.

Line184-186: specify which countries the sequences are from.

Results

Line 195: Since I recommend removing table 3, here you can complete the information regarding the

concatenated sequences.

Line 209-211: Arrange the information in such a way that Table 4 remains on a single sheet.

Line 212: We identified 52 haplotypes from the COI alignment and 66 for the ND4 with a range of 12-21

haplotypes per site.

Lines199, 218, 223: Phlebotomus argentipes

Line 231: (Fig 3)

Line: 233-234: [Anuradhapura (n=10), Balangoda (n=9), Mirigama (n=5), Medirigiriya (n=6) and Hambantota

(n=13)] (Fig 3).

Line 234-237: It would be interesting include the mutational steps between each haplotype in the figure 3.

Also indicate which haplotypes are from India and which from Sri Lanka, although the names of the localities

are indicated and a color is assigned to them, it is difficult to understand what you mean with the geographical

distribution does not coincide with the sampling localities.

Line 238: please make the haplotype matrix bigger (Fig 3, 4 and 5).

Line 250: [Anuradhapura (n=10), Balangoda (n=11), Mirigama (n=7), Medirigiriya (n=15) and Hambantota

(n=12)] (Fig 4).

Line 251: How did you determine that there were haplogroups? Please be more specific

Line 256: I understand that you analyzed the data separately (your sequences and then the sequences of

genbank). But why didn't you make a single alignment for COI?

Probably some haplotypes and haplogroups are repeating if you are looking at them individually. Did you rule

out that option?

Line 258: I also suggest including the mutational steps in the figure 4.

Line 277: Remove the references in the figure. I suggest that you explain how you obtained these haplotype

networks in the material and methods section and add the references, or describe it in detail in the results.

Discussion

Line 290: diversity (Hd> 0.5), relatively low nucleotide diversity (π= add the value).

Line 292: rather than by Indian counterparts (add the value of Hd and π for India sequences).

Line 292: the fragment length used (which gene?? Both??)

Line 296: [41]. Most of the findings in this study showed high Hd and low π (ii), which…

Was the result something you expected? Or are other categories generally reported for the genetic

diversity of the species of the genus Phlebotomus?

Line 299: as shown in?? This idea seems incomplete, please redraft. The sudden population growth that you

mention is due to human disturbance? P. argentipes is an anthropophilic species?

Line 299-303: In the line 296-299 you explain that the haplotype diversity was high and the nucleotide

diversity was low, and possibly due to sudden population growth. So why in this paragraph do you mention

low genetic diversity and bottlenecks? When previously you mention the opposite?

Please rearrange the information, the idea is not clear.

Line 304: What are the hosts of P. argentipes?, and where are their breeding sites?

Is it not an anthropophilic species of medical importance in the transmission of cutaneous

leishmaniasis? Are there wild areas in the sampled areas?

Line 305 …areas of higher epidemiological relevance [6], have been included in the

Lines 314-315: including predefined marker systems, and PCR primers known to work for sandflies

This does not seem relevant information to justify the uses of the gene ND4 is a good molecular

marker. Since several genes have these characteristics, such as COI, Cytb, ITS2, 18S etc.

Line 325: the remaining haplotypes were differentiated from the ancestor by one to three mutational steps.

Most of the mutated haplotypes…

The mutational steps data have to be included in the figures.

Line 326-327: The presence of many unique haplotypes may indicate that their populations are historically

established. (Add references and justify this argument better. Has this been recorded in other species of the

genus?

Line 327: Phlebotomus argentipes

Line 330: et al.

Line 339: Add references.

Line 342: could not

Line 345: add references

Line 349: Phlebotomus argentipes

Line 349-351: The distribution of Phlebotomus argentipes is restricted to India and Sri Lanka??

With the sequences that you analyzed, is it enough to conclude about the migratory history of the species?

Line 352-353: This work presents new insights towards understanding the genetic diversity of P. argentipes in

CL endemic areas in Sri Lanka.

How??? Explain in the discussion.

Reviewer #3: The study is very interesting, there are few studies conducted in that locality. However, caution should be taken regarding the true scope of the genes used. These mitochondrial markers are useful for discussing diversity and biogeography; however, they do not allow us to speak about ancestry. Other markers should be included, or the conclusions should be somewhat limited.

Line 87:

"The data is outdated; currently, there are 1060 sandfly species in the world (Galati, Rodriguez, 2023)."

Line 95:

The sentence is confusing: “Therefore, recent studies on leishmaniasis epidemiology have shown the augmentation of disease control due to a need for proper disease management practices”… maybe …."Therefore, recent studies on leishmaniasis epidemiology have shown an improvement in disease control due to the implementation of proper disease management practices."

Line 152:

It is important to mention the names of the primers used for PCR and the size of the amplified fragment.

Line 163:

The BLAST tool is used to confirm the identity of the sequence, not its origin.

Line 173:

It is necessary to mention the expected size of the sequence and the primers used. Generally, the fragment targeted for COI is 658 bp, but the analysis could be performed with a smaller size according to the amplified sequences. In the study, they worked with 455 bp, but the original size is not clear.

Line 280:

I suggest adding another reference; the cited reference only works with the Cyt b.

Line 347

The conclusions go beyond the results of the study. Genetic analysis using these markers (COI and ND4) does not allow confirmation of ancestry due to the inherent nature of the marker.

6. PLOS authors have the option to publish the peer review history of their article (what does this mean?). If published, this will include your full peer review and any attached files.

Reviewer #1: No

Reviewer #2: No

Reviewer #3: No

---

## [Author Response · Author response to Decision Letter 0]

11 Nov 2023

09/ 11/ 2023- PONE-D-23-20755R1 - [EMID:66daeea8c789cdfc]

We've checked your submission and before we can proceed, we need you to address the following issues:

1. Please include your tables as part of your main manuscript and remove the individual files. Please note that supplementary tables (should remain/ be uploaded) as separate "Supporting Information" files

R- We thank the editor for the comment. The copy of Table 3 submitted as an “Other” item had been removed from the system and Table 3 is included as part of the main manuscript.

25/ 09/ 2023- PONE-D-23-20755R1 - [EMID:0c2aee25263744bb]

We've checked your submission and before we can proceed, we need you to address the following issues:

1. If you are unable to obtain permission from the original copyright holder to publish these figures under the CC BY 4.0 license or if the copyright holder’s requirements are incompatible with the CC BY 4.0 license, please either i) remove the figure or ii) supply a replacement figure that complies with the CC BY 4.0 license. Please check copyright information on all replacement figures and update the figure caption with source information. If applicable, please specify in the figure caption text when a figure is similar but not identical to the original image and is therefore for illustrative purposes only.

R- Figure 1 has been replaced to comply with the aforementioned criteria. This map was originally generated in this study using ArcGIS software and thematic data from Natural Earth (public domain): http://www.naturalearthdata.com/

2. Please include a copy of Table 3 & 5 which you refer to in your paper.

R- Copy of Table 3 has been attached to the document uploading portal, as a pdf. Only four tables are included in the document and, citing of table 5 has happened as a mistake and the error has been corrected in revised manuscript line no 224.

01/ 08/ 2023- Revision required [PONE-D-23-20755] - [EMID:bc0469d5df8e1446]

R- We thank the editor for the comment. The revised manuscript has been ensured to meet the style requirements of the PLOS ONE journal.

R- We thank the editor for the comment. A statement has been included in the revised manuscript as” Permission to collect mosquito samples was obtained from the respective Medical Officer of Health (MOH) of the study areas.”

R- Thank you for your feedback. We've thoroughly reviewed and improved the manuscript's language, grammar, and spelling with the assistance of our three supervisors and language editing software, including Grammarly and Wordtune.

R- Thank you for your prompt response and guidance regarding data availability. We appreciate the opportunity to clarify our approach to data sharing. We will state that all data associated with our manuscript are fully available without restriction, and we provided the relevant accession numbers to require for access has been mentioned with supplementary materials.

5. We note that Figures 1 and 2 in your submission contain [map/satellite] images which may be copyrighted. All PLOS content is published under the Creative Commons Attribution License (CC BY 4.0), which means that the manuscript, images, and Supporting Information files will be freely available online, and any third party is permitted to access, download, copy, distribute, and use these materials in any way, even commercially, with proper attribution. For these reasons, we cannot publish previously copyrighted maps or satellite images created using proprietary data, such as Google software (Google Maps, Street View, and Earth). For more information, see our copyright guidelines: http://journals.plos.org/plosone/s/licenses-and-copyright.

R- Thank you for your feedback and guidance regarding the use of copyrighted images in Figures 1 and 2 in our manuscript. We appreciate your clarification of the publication guidelines and the necessity of adhering to the Creative Commons Attribution License (CC BY 4.0).

We have removed Figure 1 from our manuscript, as it recommended removing by all of the reviewers.

For Figure 2, we have taken the necessary steps to ensure compliance with the CC BY 4.0 license. We have regenerated Figure 2 using shape files from the publicly available, open-access and recommended database, Natural Earth (public domain): http://www.naturalearthdata.c. 

This replacement figure is in full compliance with the CC BY 4.0 license.

In the figure caption of the newly generated Figure 2, we have included the following text to acknowledge the source:

"Reprinted from Natural Earth (public domain) at [http://www.naturalearthdata.c.], under a CC BY license, with permission from Natural Earth Data”.

We believe these actions address the concerns you raised regarding copyrighted materials in our manuscript. We have ensured that all figures and content in our submission adhere to the CC BY 4.0 license and respect copyright guidelines.

Please find attached the revised Figure 2 and the updated figure caption as part of this response.

Responses to reviewers’ comments

Reviewer #1: 

The authors analyzed the genetic diversity of several populations of Phlebotomus argentipes in Sri Lanka based on two well recognized mitochondrial markers. Despite not using nuclear gene markers, many specimens were sampled for mtDNA.

Although the results increase knowledge about this important sand fly species, some adjustments need to be made to improve the article:

R- We thank reviewer 1 for his/her valuable comments, feedbacks and suggestions on our manuscript. Below changes were made accordingly in the manuscript.

i) The bibliography is out of date, some recent studies on the molecular taxonomy and populations genetics of sand flies were not included. For instance another publication on the same species and with similar objectives was recently published for Sri Lanka (https://doi.org/10.1371/journal.pone.0256819). The authors need to discuss their results in relation to this and other studies, highlighting the contributions of their findings in relation to what is already known about this species;

R- We thank the reviewer for the valuable feedback and suggestions. The revised manuscript has discussed the findings of the recent publication of Pathirage et al., 2021. We analyzed the phylogeography by using the GenBank accessions they had published. However, the alignment length was shortened after processing the alignment and the network had been narrowed down to the existing one. Hence the most important mutation points had been closed and the haplotypes had disappeared. To get a comparison to the reviewer, we attached both networks as “Annexes I” in document of “RESPONSE TO REVIEWER COMMENT”.

ii) As this is a phylogeography study, it is interesting that the authors provide a phylogenetic analysis of the analyzed sequences. This helps in interpreting the results and visualizing the clustering pattern and evolutionary relationship of the sequences. I recommend using the IQTree software, which has a user-friendly webserver (http://iqtree.cibiv.univie.ac.at/);

R- Thank you for your valuable feedback and suggestion regarding the inclusion of a phylogenetic analysis in our phylogeography study.

We appreciate your thoughtful recommendation, and we agree that phylogenetic analysis can provide valuable insights into the clustering patterns and evolutionary relationships of the analyzed sequences. In our study, we have chosen to employ a haplotype network approach as it aligns well with our research objectives to expand the clustering in relevant to phylogeography and the nature of our dataset. Haplotype network analysis has proven to be a powerful tool for visualizing genetic relationships among closely related sequences, allowing us to examine the genealogy and population structure in a manner that complements the goals of our phylogeography study. 

Our primary goal is to provide a clear and concise presentation of our findings while maintaining the focus on the core objectives of our research, which is centered on phylogeographic patterns and population dynamics. Given the extensive nature of our dataset and the unique characteristics of each dataset, the inclusion of multiple phylogenetic trees could potentially divert the reader's attention from our central message.

We hope that you understand our reasoning for this decision and trust that our chosen approach aligns with the goals and scope of our study. We remain open to further suggestions or concerns you may have and are committed to ensuring the clarity and quality of our manuscript.

iii) Genetic structure indices (such as the Fst) can improve the understanding on the molecular divergence (or the lack of) between the analyzed populations.

R- We have already submitted another manuscript that comprehensively studied the population structure and the demographic history of P.argentipes in Sri Lanka. Therefore the recent manuscript has focused only on the genetic diversity and phylogeography of the P.argentipes in Sri Lanka.

Other minor corrections and suggestions are in the attached pdf file.

1. Line 37: The name of the disease must be written with lowercase initials, please correct it in the rest of the manuscript.

R- This has been corrected in the revised manuscript in line no 38, 97-98.

2. Line 50: Cytochrome c oxidase subunit I

R- “Cytochrome Oxidase I” has been changed to “Cytochrome c oxidase subunit I” in the revised manuscript in line no 51.

3. Line 87: this number is quite out of date, please check the following reference https://doi.org/10.1007/s13744-023-01030-8

R- According to the reference the number of species count, has been changed in revised manuscript line no 105.

4. Line 88: main

R- This has been corrected to “main” in the revised manuscript line no 106.

5. Line 95: the inclusion of this figure in the introduction is unnecessary. Alternatively, disease endemicity information may be included in figure 2

R- This figure (Figure 1) has been removed from the revised manuscript. The disease epidermicity data has been included to the text in revised manuscript line no 101- 104.

6. Line 108: give full species names at the first mention. Need to standardize the abbreviation of genera, check the following reference: https://doi.org/10.3157/0013 872X(2007)118[351:APOGAS]2.0.CO;2

R- The full species name has been included in the revised manuscript line no 1129.

7. Line 131: This section does not make it clear how the samples were processed. Here, it needs to be informed which parts of the specimens were used for morphological identification (were they mounted on slides and deposited in a collection?), and which were stored for further DNA extraction.

R- These unclear points that were included while previous writing was clearly mentioned in the revised manuscript line no 159- 167, 175- 179

8. Line 138: Do not abbreviate the genus name at the beginning of sentences

R- This has been corrected in the revised manuscript.

9. Line 151: What experiments were the heads used in?

R- The head parts were preserved to perform comprehensive observation of head-based morphometric assessment studies.

10. Line 160: This table was not referenced in the previous paragraph. It can be moved or referenced where it best fits

R- Table 2 has been integrated with Table 3 in the revised manuscript line no 228 and the initial Table 2 legend was removed and reorganized according to all reviewers’ comments.

11. Line 161: The description of these two COI datasets should be more clear and explicit in the text.

R- This has been clearly mentioned in the revised manuscript line no 197- 199. To remove the contradiction relevant to this data set explanation, some sentences were in previous manuscript having been removed from the revised manuscript.

12. Line 194- 195: In table 3, this value (121) does not exist, since you put the number of haplotypes

R- Table 3 has been rearranged by including the mentioned data by the reviewer and included as table 2 in the revised manuscript line no 228.

13. Line 201: Was this the exact size of the alignment used? If not, provide the exact values

R- Exact size of the PCR products and alignments has been mentioned in revised manuscript line no 232 and 197- 199 respectively.

Reviewer #2: 

The manuscript "Genetic diversity and phylogeography of Phlebotomus argentipes (Diptera: Psychodidae), using COI and ND4 mitochondrial gene sequences” provides relevant information on the genetic diversity and phylogeopgraphic distribution of the sand fly Phlebotomus argentipes from Sri Lanka. In addition, it contributes to new sequences of mitochondrial markers COI and ND4 for Phlebotomus argentipes. However, some corrections/suggestions are necessary, and are detailed in the attached file. I suggested that you do a language review, there are several errors and it is not very clear in some sentences. Despite this, I considered that the authors have valuable information that can be published if they make improvements to their manuscript.

R- We thank reviewer 2 for his/her valuable comments, feedbacks and suggestions on our manuscript. Below changes were made accordingly in the manuscript. Corrections have been done in revised manuscript.

Title: Lines1-3: I suggest added Phlebotominae in the title: Genetic diversity and phylogeography of Phlebotomus argentipes (Diptera: Psychodidae, Phlebotominae) using COI and ND4 mitochondrial gene sequences. Delete the point at the end of the title.

R- “Phlebotominae” has been added to the title and the point has been removed from the title in revised manuscript lines no 2.

Key words: Cutaneous Leishmaniasis; Sandfly; Phlebotomus argentipes, Sri Lanka; Haplotypes, phylogeny, Do not use the same words as in the title. Find complementary words.

R- We have revised the text to include additional keywords and complementary terms to avoid repetition and enhance the clarity of the content.“Genetic variation, Mitochondrial markers, Insect vectors, Sandfly species, Evolutionary history, Leishmania transmission”

Abstract

Line 49: A total of 159 P. argentipes specimens, were collected and identified from five cutaneous leishmaniasis endemic areas.

R- This correction has been made in the revised manuscript in line no 50.

Lines 50: Cytochrome Oxidase subunit I

R- This has been corrected in the revised manuscript in line no 51.

Introduction

The introduction needs to be more specific for the study area, and to the sand fly species to be analysed. There is a lot of information that is not relevant to this job. Please provide specific information (background) on the topic you are going to develop.

R- The introduction has been rewritten in the revised manuscript.

Line 87: Out of 1,060 (Please update the number of species) sand fly species in the world, Phlebotomus argentipes Annandale and Brunetti (year???) is one of the major vectors for Visceral Leishmaniasis (VL) and Cutaneous Leishmaniasis (CL) in Sri Lanka???(or where?? specify). I share this reference where the number of species worldwide is updated: Galati E, Rodrigues B. A review of historical Phlebotominae taxonomy (Diptera: Psychodidae). Neotrop Entomol. 2023; 1–21.doi:10.1007/s13744-023-01030-8

R- The changes have been made in the revised manuscript lines no 105-106, 101- 104.

Line 89: Being CL the most common form, annually reporting 0.7-1.2 90 million cases worldwide [4], and caused by nearly 20 species of obligate intracellular protozoa Leishmania sp. (Euglenozoa: Trypanosomatidae) [5]. I agree that cutaneous leishmaniasis is the most common clinical form worldwide. But this sentence is not so relevant in this part of the manuscript. I suggest that you to rewrite this paragraph.

R- We thank the reviewer for the valuable feedback and suggestions. The paragraph is reorganized and the mentioned sentence has been removed.

Lines 91-95: In Sri Lanka, around 1,991 annual CL cases have been recorded mainly in endemic areas of the North Central Province (Anuradhapura District and Polonnaruwa District) and Southern Province (Matara and Hambantota districts) in the country (Fig. 1) [6-7]. Please change Figure to Fig throughout the text, considering the guidelines of the journal.

R- Sentence has been corrected as mentioned in the revised manuscript line no 101- 104.

Lines 98-99: are believed to be high in the country. What do you mean with “are believed”? I suggest you be more specific. Please only use facts (references) not assumptions.

R- The correction has been done in the revised manuscript line no 120- 124.

Line 101: host communities, host species turnover??. Do you mean to blood meal preferences with this?

R- The correction has been done in revised manuscript line no 124-125.

Line 101: vectorial capacity and competence. Do you considered the vectorial competence too??

R- Mentioned sentence has been removed from the revised manuscript.

Lines 103-105: I don’t understand why you include a map with the incidence of cutaneous leishmaniasis if your manuscript focuses on the genetic diversity of sand fly Phlebotomus argentipes? Does the distribution of Phlebotomus argentipes coincides with the sites where the cases have been reported? In this case is better including the distribution of the sand fly. I suggest that this map of cutaneous leishmaniasis cases send as supplementary information. Only include a map indicating the sites where you collected for your study (in this case the Figure 2 should be the Figure 1).

R- Mentioned figure (Fig 1) has been removed from the revised manuscript. The disease epidermicity data has been included in the text in the revised manuscript line no 101- 104.

Line 103: Fig 1. The distribution of…

R- The figure has been removed from the revised manuscript.

Lines 106-107: add the complete name of the genes since it is the first time they appear. [microsatellites, the Internal Transcribed Spacer 2 (ITS2)] and mitochondrial markers [cytochrome b (cytb), cytochrome oxidase subunit 1 (COI) and NADH Dehydrogenase 4 (ND4)] have been used [11-17], to determine the genetic diversity and phylogeography of different sand fly species such as P. papatasi, P. perniciosus and Lu. Longipalpis.

R- Sentence has been reorganized and corrected in revised manuscript line no 126- 127. Complete name of COI and ND4 genes have been abbreviated in abstract line no 51-52 in the revised manuscript.

Lines 111-112: Thus, mitochondrial genes have been used in evolutionary studies of vector species including phlebotomine sandflies. (It is necessary including a reference).

R- New references have been added to the revised manuscript line no 141.

Lines 112-114: However, most studies have used few mitochondrial genes to estimate the phylogeography of sand fly species.

R- We thank the reviewer for the valuable feedback and the correction has been done in revised manuscript line no 122-124.

Lines 114-118: In the present study, extended fragments of the COI gene, exceeding the lengths employed in previous investigations of P. argentipes, meanwhile the ND4 fragments characterized by elevated mutation rates, were utilized to explore the phylogeography and genetic diversity of P.argentipes in Sri Lanka. Since in the previous paragraph I suggested putting the full name of the genes, in this paragraph you can abbreviated.

R- The sentence has been corrected in the revised manuscript line no 141- 144.

Line 130: Material and Methods

R- This has been corrected in the revised manuscript line no 153.

Line 132: In those study areas there were no other species of sandflies?

R- We thank the reviewer for the valuable feedback. Few Sergentomyia sp. individuals had been identified from some locations and the current study had been focused on P.argentipes population only.

Line 132: randomly??, Do you mean that during the three years of the sampling, you always placed the traps in different places? Please explain why were the traps not placed in the same place during sampling? What would be the explanation for the low abundance of this species during the three years of sampling? Since is a vector, I consider that for a sampling of three years, 159 specimens are very few specimens. If you collected more specimens but only used 159 for the molecular analysis, you should specify it.

R- Mentioned corrections have been done in the revised manuscript line no 159. We placed traps in the same collection sites and repeatedly and only focused on P.argentipes species. Very few numbers of Sentomyierga sp and Culicoides sp were also collected. Only 159 P.argentipes individuals from the total collection were used for genetic analysis.

Lines 133-1334: from five localities in Sri Lanka from March 2018 to March 2020 (Table 1).

R- This has been corrected in the revised manuscript line no 155- 156.

Line 135: (Fig 2, Table 1).

R- This has been corrected in the revised manuscript line no 158.

Line 138: for further analysis. Phlebotomus argentipes specimens…

R- This has been corrected in the revised manuscript line no 164.

Line 140: I suggest that in Table 1 you also add the Genbank accession numbers generated in this study for COI and NAD4 genes.

R- Table 1 has been modified by including the accession numbers as you mentioned. Changed table can be found in the revised manuscript line no 169.

Line 144: Fig 2. Geographic distribution of Phlebotomus argentipes samples analysed. The sampling locations of this study are marked with a triangle, and the GenBank sequences retrieved for mitochondrial DNA fragments from Sri Lanka with a square, and those from India with a circle. I suggest that you make the letter and symbols bigger for better visualization of the figure.

R- This figure has been regenerated with the requested changes in the revised manuscript line no 173.

Line 149: How did you perform the morphological identification if you used the complete body for extraction and the head for other experiments? For the morphological identification of Phlebotomus argentipes it is not necessary to visualize the cibarium or the spermatheca? Please justify how you made the morphological identification and if you made assemblies, what technique you followed and where the specimens are kept?

R- We thank the reviewer for the comment. After finishing the identification process through the microscopic image processing system by using the whole body of the insect, identified P.argentipes individuals were preserved with 70% ethanol in separate tubes prior to DNA extraction. Sorry about the ambiguity, and this section has been corrected in the revised manuscript line no 156- 160, 175- 178.

Line 160: Since the information that is in Table 1 is displayed in Figure 2, I consider it appropriate to join table 1 and 2 and only keep the relevant information, to avoid the information being repetitive and confusing. Only the data that you generated should be included in the table. The data that you downloaded from GenBank it is written in the paragraph (Line184-186), it is not necessary to repeat the information.

R- The table 1 has been modified in revised manuscript line no 169 and the regenerated figure 2 also has been included in revised manuscript line no 173 as figure 1.

Lines 173-174: The information in Table 3 can be described in the paragraph. It is not necessary the table to describe this information. Please remove the table.

R- Table 3 has been removed and the information has been described in the revised manuscript line no 198- 199.

Line 177: specify what software you used to perform this procedure.

R- We thank the reviewer for the comment. “DnaSP software” has been mentioned in the revised manuscript line no 208.

Lines 181-182: I do not understand, what you mean with MZ882190 to MZ882255. Please be more specific, because it's confusing this way. Please do not repeat the information, is better if you refer to the table or figure.

R- Table 1 has been regenerated by including the suggested information and cited in the revised manuscript line no 169.

Line184-186: specify which countries the sequences are from.

R- Sequences deposited by Indian studies were used for the study and the correction on the text has been made in the revised manuscript line no 216.

Results

Line 195: Since I recommend removing table 3, here you can complete the information regarding the concatenated sequences.

R- Information regarding the concatenated analysis, has been included in Table 2 in the revised manuscript line no 228.

Line 209-211: Arrange the information in such a way that Table 4 remains on a single sheet.

R- The table has been fixed for a single sheet in the revised manuscript line no 239.

Line 212: We identified 52 haplotypes from the COI alignment and 66 for the ND4 with a range of 12-21 haplotypes per site.

R- The correction has been made in the revised manuscript line no 240.

Lines199, 218, 223: Phlebotomus argentipes

R- All corrected in revised manuscript.

Line 231: (Fig 3)

R- The correction has been done in the revised manuscript line no 273.

Line: 233-234: [Anuradhapura (n=10), Balangoda (n=9), Mirigama (n=5), Medirigiriya (n=6) and Hambantota (n=13)] (Fig 3).

R- The correction has been made in the revised manuscript line no 264.

Line 234-237: It would be interesting include the mutational steps between each haplotype in the figure 3. Also indicate which haplotypes are from India and which from Sri Lanka, although the names of the localities are indicated and a color is assigned to them, it is difficult to understand what you mean with the geographical distribution does not coincide with the sampling localities.

R- We thank the reviewer for the valuable feedback and suggestions. Mutational steps of the haplotypes were included in the S1 table and representing the mutational points between each haplotype has been created unclear nature for the figure, hence the network with mutational steps has been inserted as the S1 figure. Haplotype IDs from each population set have also been included in S6 table. Network trees in the current study, had been displayed as star-shape which indicates population expansion, rather than a subdivision. So In Sri Lanka, the P.aregentipes population has been identified as homogenized and not coincidentally with geographical distribution.

We deeply regret the ambiguity here and have included the mentioned facts by the reviewer, in the revised manuscript line no 267- 269.

Line 238: please make the haplotype matrix bigger (Fig 3, 4 and 5).

R- The font size of the haplotype IDs has been increased in the revised manuscript.

Line 250: [Anuradhapura (n=10), Balangoda (n=11), Mirigama (n=7), Medirigiriya (n=15) and Hambantota (n=12)] (Fig 4).

R- The correction has been made in the revised manuscript line no 282- 283.

Line 251: How did you determine that there were haplogroups? Please be more specific

R- Haplogroups have been identified based on the clustering of closely related haplotypes that share specific genetic mutations in the haplotype network. The correction has been done in manuscript line no 265.

Line 256: I understand that you analyzed the data separately (your sequences and then the sequences of genbank). But why didn't you make a single alignment for COI? Probably some haplotypes and haplogroups are repeating if you are looking at them individually. Did you rule out that option?

R- The study had considered all COI sequences in “COI regional alignment” both COI study sequences and Genbank accessions were there. 

Haplotype topology was done simultaneously for both COI alignments separately, and it was done by observing the segregation points of each haplotype point for both alignments separately. Haplotypes from HC1 to HC52 were initially done with their nomenclature from the "COI study alignment", and the haplotypes from HC53 to HC56 were the newly generated haplotypes in "COI regional alignment".

To minimize the contradiction, the correction has been done in the revised manuscript line no 197- 199.

Line 258: I also suggest including the mutational steps in the figure 4.

R- Mutational steps between each haplotype have been included in Fig S2 in the revised manuscript.

Line 277: Remove the references in the figure. I suggest that you explain how you obtained these haplotype networks in the material and methods section and add the references, or describe it in detail in the results.

R- We thank the reviewer for the valuable feedback and the correction has been done in the revised manuscript line no 312.

Discussion

Line 290: diversity (Hd> 0.5), relatively low nucleotide diversity (π= add the value).

R- Relevant values have been included in the revised manuscript line no 328.

Line 292: rather than by Indian counterparts (add the value of Hd and π for India sequences).

R- By including the relevant values in there, correction has been done in the revised manuscript line no 331.

Line 292: the fragment length used (which gene?? Both??)

R- The length of the fragment COI has been mentioned in the revised manuscript line no 331.

Line 296: [41]. Most of the findings in this study showed high Hd and low π (ii), which… Was the result something you expected? Or are other categories generally reported for the genetic diversity of the species of the genus Phlebotomus?

R- We thank the reviewer for the valuable feedback. We appreciate your query about the consistency of our results with other studies within the genus Phlebotomus. In our study, we did find that most of our findings demonstrated high haplotype diversity (Hd) and low nucleotide diversity (π) falling under category (ii) While these outcomes align with certain previous studies based on most of the sandfly species. It is noteworthy that the available local previous publication on this specific species yielded similar results, albeit with a different interpretation. The correction has been done in the revised manuscript line no 343- 346.

Line 299: as shown in?? This idea seems incomplete, please redraft. The sudden population growth that you mention is due to human disturbance? P. argentipes is an anthropophilic species?

R- As an anthropophilic vector, P.argentipes population size can be influenced by human activities such as habitat modification, urbanization, Insecticide usage, the environmental changes also impact. Changes to the revised manuscript (line no 343- 359) have been done.

Line 299-303: In the line 296-299 you explain that the haplotype diversity was high and the nucleotide diversity was low, and possibly due to sudden population growth. So why in this paragraph do you mention low genetic diversity and bottlenecks? When previously you mention the opposite? Please rearrange the information, the idea is not clear.

R- This section has been rewritten and included in revised manuscript lines no 343- 359).

Line 304: What are the hosts of P. argentipes?, and where are their breeding sites? Is it not an anthropophilic species of medical importance in the transmission of cutaneous leishmaniasis? Are there wild areas in the sampled areas?

R- P.argentipes is an anthropophilic and opportunistic zoophilic vector species for leishmaniasis. The section has been rewritten and included in the revised manuscript lines no 343- 359.

Line 305 …areas of higher epidemiological relevance [6], have been included in the

R- Mentioned correction has been done in the revised manuscript line no 359.

Lines 314-315: including predefined marker systems, and PCR primers known to work for sandflies This does not seem relevant information to justify the uses of the gene ND4 is a good molecular marker. Since several genes have these characteristics, such as COI, Cytb, ITS2, 18S etc.

R- Justification relevant to the user success of ND4 marker has been included in the revised manuscript line no 369- 384.

Line 325: the remaining haplotypes were differentiated from the ancestor by one to three mutational steps. Most of the mutated haplotypes… The mutational steps data have to be included in the figures.

R- The correction has been done in the revised manuscript line no 413. Mutational steps have been mentioned in the S1 table and the requested modified figure has been submitted as S1 and S2 figures.

Line 326-327: The presence of many unique haplotypes may indicate that their populations are historically established. (Add references and justify this argument better. Has this been recorded in other species of the genus?

R- Reference has been added to the revised manuscript line no 394- 400.

Line 327: Phlebotomus argentipes

R- The correction has been made in the revised manuscript line no 397

 Line 330: et al.

R- Correction has been done in the revised manuscript line no 400.

Line 339: Add references.

R- Reference has been added to the revised manuscript line no 410.

Line 342: could not

R- The correction has been made in the revised manuscript line no 426.

Line 345: add references

R- Reference has been added to the revised manuscript line no 419.

Line 349: Phlebotomus argentipes

R- The section has been rearranged in the revised manuscript.

Line 349-351: The distribution of Phlebotomus argentipes is restricted to India and Sri Lanka?? With the sequences that you analyzed, is it enough to conclude about the migratory history of the species?

R- We thank the reviewer for the valuable feedback and suggestions. P.argentipes can be found in a few Asian countries like Nepal, Pakistan, Thailand, Myanmar etc. India and Sri Lanka had been identified as P. argentipes prevalence countries with a focus on leishmania epidermic. The correction has been done in the revised manuscript line no 426 by removing the word “migratory” which is an uncertain fact.

Line 352-353: This work presents new insights towards understanding the genetic diversity of P. argentipes in CL endemic areas in Sri Lanka. How??? Explain in the discussion.

R- Explanation has been included in the revised manuscript line no 440- 444.

Reviewer #3: 

The study is very interesting, there are few studies conducted in that locality. However, caution should be taken regarding the true scope of the genes used. These mitochondrial markers are useful for discussing diversity and biogeography; however, they do not allow us to speak about ancestry. Other markers should be included, or the conclusions should be somewhat limited.

R- We thank reviewer 3 for his/her valuable comments, feedbacks and suggestions on our manuscript. Below changes were made accordingly in the manuscript

Line 87: "The data is outdated; currently, there are 1060 sandfly species in the world (Galati, Rodriguez, 2023)."

R- According to the reference, the number has been changed in the revised manuscript line no 105.

Line 95: The sentence is confusing: “Therefore, recent studies on leishmaniasis epidemiology have shown the augmentation of disease control due to a need for proper disease management practices”… maybe …."Therefore, recent studies on leishmaniasis epidemiology have shown an improvement in disease control due to the implementation of proper disease management practices."

R- The sentence has been corrected in the revised manuscript line no 104 to extract the exact meaning.

Line 152: It is important to mention the names of the primers used for PCR and the size of the amplified fragment.

R- Primer IDs and the sequences have been included in the revised manuscript line no 180-181, 185-186. The size of the amplified products was mentioned in the results (line no 232).

Line 163: The BLAST tool is used to confirm the identity of the sequence, not its origin.

R- The word “origin” has been removed from the sentence in revised manuscript (line no 194).

Line 173: It is necessary to mention the expected size of the sequence and the primers used. Generally, the fragment targeted for COI is 658 bp, but the analysis could be performed with a smaller size according to the amplified sequences. In the study, they worked with 455 bp, but the original size is not clear.

R- Amplified PCR product sizes were 500bp and 800bp for COI and ND4 respectively. But at the alignment preparation, sequences were trimmed to remove primer binding sites and remain only the protein coding piece only. This explanation has been included to the revised manuscript line no 232.

Line 280: I suggest adding another reference; the cited reference only works with the Cyt b.

R- References related to other genes are also cited in the revised manuscript line no 318.

Line 347: The conclusions go beyond the results of the study. Genetic analysis using these markers (COI and ND4) does not allow confirmation of ancestry due to the inherent nature of the marker.

R- We thank the reviewer for the valuable feedback and suggestions. The conclusion of the revised manuscript has been rearranged (line no 421- 444).

---

## [Decision Letter · Decision Letter 1]

30 Nov 2023

PONE-D-23-20755R1Genetic diversity and phylogeography of Phlebotomus argentipes (Diptera: Psychodidae, Phlebotominae), using COI and ND4 mitochondrial gene sequencesPLOS ONE

Dear Dr. De Silva,

Thank you for submitting your manuscript to PLOS ONE. After careful consideration, we feel that it has merit but does not fully meet PLOS ONE’s publication criteria as it currently stands. Therefore, we invite you to submit a revised version of the manuscript that addresses the points raised during the review process.

We look forward to receiving your revised manuscript.

Kind regards,

Felipe Dutra-Rêgo, PhD

Academic Editor

PLOS ONE

Journal Requirements:

Additional Editor Comments:

Please, check the reviewers' comments. Several minor lacunas need to be corrected prior publication.

Reviewers' comments:

Reviewer's Responses to Questions

**Comments to the Author**

1. If the authors have adequately addressed your comments raised in a previous round of review and you feel that this manuscript is now acceptable for publication, you may indicate that here to bypass the “Comments to the Author” section, enter your conflict of interest statement in the “Confidential to Editor” section, and submit your "Accept" recommendation.

Reviewer #1: All comments have been addressed

Reviewer #2: All comments have been addressed

Reviewer #3: (No Response)

2. Is the manuscript technically sound, and do the data support the conclusions?

Reviewer #1: Yes

Reviewer #2: Yes

Reviewer #3: Yes

3. Has the statistical analysis been performed appropriately and rigorously? 

Reviewer #1: Yes

Reviewer #2: Yes

Reviewer #3: N/A

4. Have the authors made all data underlying the findings in their manuscript fully available?

Reviewer #1: Yes

Reviewer #2: Yes

Reviewer #3: Yes

5. Is the manuscript presented in an intelligible fashion and written in standard English?

Reviewer #1: Yes

Reviewer #2: Yes

Reviewer #3: Yes

6. Review Comments to the Author

Reviewer #1: Dear Authors,

Thank you for providing a detailed response letter for my suggestions and comments. I have no longer major comments about your manuscript.

Best regards.

Reviewer #2: The manuscript "Genetic diversity and phylogeography of Phlebotomus argentipes (Diptera: Psychodidae, Phlebotominae), using COI and ND4 mitochondrial gene sequences” provides relevant information on the genetic diversity. I thank the authors for considering my observations and including them in their work. I extend congratulations on your research achievements. Although the article has improved a lot and I believe it can be published, some spelling errors should be corrected first by the authors. I have attached my final comments. Best regards.

Line 52: delete extra parenthesis (ND4)

Line 107: P. argentipes

Line 127: delete extra parenthesis (ITS 2), and change parenthesis by brackets [cytochrome b (Cyt b), COI, and ND4],

Line 160-161: A total of 159 specimens of P. argentipes were used for the present analysis.

Line 166-167: Identified sandfly specimens were preserved in 70% ethanol before extracting DNA.

Line 176: please change individuals by “specimens”

Line 180-182: each primer C1J (5'- GGA GGA TTT GGA AAT TGA TTA GTT C-3') and C1N (5'- CCC GGT AAA ATT AAA ATA TAA ACT TC-3'),

Line 185-186: each primer ND4ar (5’- AA(A/G) GCT CAT GTT GAA GC-3’) and ND4c (5’-ATT TAA AGG (T/C)AA TCA ATG TAA-3’),

For these two comments, I suggest you put the oligos names first and then the sequence (between parentheses), as well as separate them by triplets.

Line 203-204: …9,494 bp regions respectively, against the reference sequence (KR349298.1) of P. papatasi complete mitochondrial genome [39] as no P. argentipes reference sequences available in databases.

Line 214-217: The unique haplotypes for the Asian region (regional analysis) were analyzed using our COI sequences and 19 COI sequences previously deposited in the GenBank repository from India [41] and Sri Lanka (KC791432 to KC791437, HQ541166, HQ585366 to HQ585373, and KT428789 to KT478792) [10].

Line 225: please change individuals by “specimens”

Line 272-273: Circle size indicates the haplotype frequency, and the circle color the geographical location. Haplotype labels are written next to the corresponding circles.

Line 295-296: Circle size and color indicate the frequency and geographical location of the haplotypes respectively. Haplotype labels are written next to the corresponding circles, and the red circles indicate median vectors.

Line 312-315: Circle size and color indicate haplotype frequency and geographical location respectively. Haplotype labels are written next to the corresponding circles: (A) GenBank retrieved COI sequences/ regional alignment, (B) haplotypes derived from a previous study data set.

Line 365-368: The study of genetic markers such as the ND4 gene has allowed for a finer resolution in the analysis of genetic variability within P. argentipes populations. Identifying numerous unique haplotypes, this research contributes to understanding the intricate genetic landscape of this sandfly species in CL endemic regions of Sri Lanka.

Line 381: cryptic species

Reviewer #3: Line 96

The Leishmania protozoa in their intracellular form are not considered flagellates, as the flagellum in this form is almost imperceptible; therefore, the phrase becomes confusing. I suggest replacing it with "caused by protozoan parasites of the genus Leishmania.

7. PLOS authors have the option to publish the peer review history of their article (what does this mean?). If published, this will include your full peer review and any attached files.

Reviewer #1: No

Reviewer #2: No

Reviewer #3: No

---

## [Author Response · Author response to Decision Letter 1]

8 Dec 2023

PLOS ONE Decision: Revision required [PONE-D-23-20755R1] - [EMID:ae3de9bee523e054]

Journal requirements

R- We thank the editor for their prompt review and valuable feedback. We have thoroughly reviewed the reference list for the manuscript, and we can confirm that none of the cited articles have been retracted. We have carefully examined each reference, and all of them are accurate and up-to-date.

Additional Editor Comments:

Please, check the reviewers' comments. Several minor lacunas need to be corrected prior publication.

R- Corrections has been done accordingly

Reviewer #1: 

Dear Authors, Thank you for providing a detailed response letter for my suggestions and comments. I have no longer major comments about your manuscript.

Best regards.

R- We express our gratitude to the reviewer for their invaluable feedback, which has significantly contributed to the enhancement of this work.

Reviewer #2: 

The manuscript "Genetic diversity and phylogeography of Phlebotomus argentipes (Diptera: Psychodidae, Phlebotominae), using COI and ND4 mitochondrial gene sequences” provides relevant information on the genetic diversity. I thank the authors for considering my observations and including them in their work. I extend congratulations on your research achievements. Although the article has improved a lot and I believe it can be published, some spelling errors should be corrected first by the authors. I have attached my final comments. Best regards.

R- We express our gratitude to Reviewer 2 for providing valuable comments, feedback, and suggestions on our manuscript. These inputs have significantly contributed to the improvement of our work. We have implemented the suggested corrections as advised.

Line 52: delete extra parenthesis (ND4)

R- Extra parenthesis has been removed in the revised manuscript line no 52.

Line 107: P. argentipes

R- Mentioned correction has been done in the revised manuscript line no 107.

Line 127: delete extra parenthesis (ITS 2), and change parenthesis by brackets [cytochrome b (Cyt b), COI, and ND4],

R- Extra parenthesis has been removed in line no 127 and the mentioned brackets have been included in same line in the revised manuscript.

Line 160-161: A total of 159 specimens of P. argentipes were used for the present analysis.

R- Mentioned correction has been done in the revised manuscript line no 160.

Line 166-167: Identified sandfly specimens were preserved in 70% ethanol before extracting DNA.

R- Mentioned correction has been done in the revised manuscript line numbers 165- 166.

Line 176: please change individuals by “specimens”

R- The word “individuals” has been replaced with “specimens” in the whole document with relevance.

Line 180-182: each primer C1J (5'- GGA GGA TTT GGA AAT TGA TTA GTT C-3') and C1N (5'- CCC GGT AAA ATT AAA ATA TAA ACT TC-3'),

R- Mentioned correction has been done in the revised manuscript line numbers 179- 181.

Line 185-186: each primer ND4ar (5’- AA(A/G) GCT CAT GTT GAA GC-3’) and ND4c (5’-ATT TAA AGG (T/C)AA TCA ATG TAA-3’), For these two comments, I suggest you put the oligos names first and then the sequence (between parentheses), as well as separate them by triplets.

R- R- Mentioned correction has been done in the revised manuscript line numbers 184- 186.

Line 203-204: …9,494 bp regions respectively, against the reference sequence (KR349298.1) of P. papatasi complete mitochondrial genome [39] as no P. argentipes reference sequences available in databases.

R- The sentence has been corrected in revised manuscript line numbers 202- 204.

Line 214-217: The unique haplotypes for the Asian region (regional analysis) were analyzed using our COI sequences and 19 COI sequences previously deposited in the GenBank repository from India [41] and Sri Lanka (KC791432 to KC791437, HQ541166, HQ585366 to HQ585373, and KT428789 to KT478792) [10].

R- The sentence has been corrected in revised manuscript line numbers 213- 216.

Line 225: please change individuals by “specimens”

R- Word has replaced with “specimens” in revised manuscript line no 223.

Line 272-273: Circle size indicates the haplotype frequency, and the circle color the geographical location. Haplotype labels are written next to the corresponding circles.

R- The sentence has been corrected in the revised manuscript lines 270- 271.

Line 295-296: Circle size and color indicate the frequency and geographical location of the haplotypes respectively. Haplotype labels are written next to the corresponding circles, and the red circles indicate median vectors.

R- The sentence has been corrected in the revised manuscript lines 292- 294.

Line 312-315: Circle size and color indicate haplotype frequency and geographical location respectively. Haplotype labels are written next to the corresponding circles: (A) GenBank retrieved COI sequences/ regional alignment, (B) haplotypes derived from a previous study data set.

R- The sentence has been corrected in the revised manuscript lines 310- 312.

Line 365-368: The study of genetic markers such as the ND4 gene has allowed for a finer resolution in the analysis of genetic variability within P. argentipes populations. Identifying numerous unique haplotypes, this research contributes to understanding the intricate genetic landscape of this sandfly species in CL endemic regions of Sri Lanka.

R- Mentioned correction has been done in the revised manuscript line numbers 362- 365.

Line 381: cryptic species

R- The sentence has been corrected in the revised manuscript lines 378.

Reviewer #3: 

Line 96 The Leishmania protozoa in their intracellular form are not considered flagellates, as the flagellum in this form is almost imperceptible; therefore, the phrase becomes confusing. I suggest replacing it with "caused by protozoan parasites of the genus Leishmania.

R- We thank the reviewer for the valuable feedback and the correction has been done in the revised manuscript line no 96.

---

## [Editor Report · Decision Letter 2]

10 Dec 2023

Genetic diversity and phylogeography of Phlebotomus argentipes (Diptera: Psychodidae, Phlebotominae), using COI and ND4 mitochondrial gene sequences

PONE-D-23-20755R2

Dear Dr. De Silva,

We’re pleased to inform you that your manuscript has been judged scientifically suitable for publication and will be formally accepted for publication once it meets all outstanding technical requirements.

Kind regards,

Felipe Dutra-Rêgo, PhD

Academic Editor

PLOS ONE
---

## [Editor Report · Acceptance letter]

15 Dec 2023

PONE-D-23-20755R2 

PLOS ONE

Dear Dr. De Silva, 

I'm pleased to inform you that your manuscript has been deemed suitable for publication in PLOS ONE. Congratulations! Your manuscript is now being handed over to our production team.

Kind regards, 

on behalf of

Dr. Felipe Dutra-Rêgo 

Academic Editor

PLOS ONE